

# Helicopter-borne observations of the continental background aerosol in combination with remote sensing and ground-based measurements.

Sebastian Düsing[1], Birgit Wehner[1], Patric Seifert[1], Albert Ansmann[1], Holger Baars[1], Florian Ditas[1,2], Silvia Henning[1], Nan Ma[1], Laurent Poulain[1], Holger Siebert[1], Alfred Wiedensohler[1] and Andreas Macke[1]

[1]Leibniz Institute for Tropospheric Research (TROPOS), 04318 Leipzig, Germany
[2]Multiphase Chemistry Department, Max Planck Institute for Chemistry, P.O. Box 3060, 55020, Mainz, Germany

*Correspondence to:* Sebastian Düsing (duesing@tropos.de)

**Abstract**. This study presents vertical profiles up to a height of 2300 m a.s.l. of aerosol microphysical and optical properties and cloud condensation nuclei (CCN). Corresponding data have been measured during a field campaign as part of the High-Definition

Clouds and Precipitation for Advancing Climate Prediction (HD(CP)²) Observational Prototype Experiments (HOPE), which took place at Melpitz, Germany from September 9 to 29, 2013.

The helicopter-borne payload ACTOS (Airborne Cloud and Turbulence Observation System) was used to determine the aerosol particle number size distribution (PNSD), the number concentrations of aerosol particles (PNC) and cloud condensation nuclei (CCN) (CCN-NC), the ambient relative humidity (*RH*), and temperature (*T*). Simultaneous measurements on ground provided a

holistic view on aerosol microphysical properties such as the PNSD, the chemical composition and the CCN-NC. Additional measurements of a 3+2 wavelength polarization lidar system (Polly$^{XT}$) provided profiles of the aerosol particle light backscatter coefficient ($\sigma_{bsc}$) for three wavelengths (355, 532 and 1064 nm). From profiles of $\sigma_{bsc}$ profiles of the aerosol particle light extinction coefficient ($\sigma_{ext}$) were determined using the extinction-to-backscatter ratio. Furthermore, CCN-NC profiles were estimated on basis of the lidar-measurements.

Ambient state optical properties of aerosol particles were derived on the basis of airborne in situ measurements of ACTOS (PNSD) and in situ measurements on ground (chemical aerosol characterization) using Mie-theory.

On the basis of ground-based and airborne measurements, this work investigates the representativeness of ground-based aerosol microphysical properties for the boundary layer for two case-studies. The PNSD measurements on ground showed a good agreement with the measurements provided with ACTOS for lower altitudes. The ground-based measurements of PNC and CCN-NC are

representative for the PBL when the PBL is well mixed. Locally isolated new particle formation events on ground or at the top of the PBL led to vertical variability in the here presented cases and ground- based measurements are not representative for the PBL. Furthermore, the lidar-based estimates of CCN-NC profiles were compared with the airborne in situ measurements of ACTOS. This comparison showed good agreements within the uncertainty range.

Finally, this work provides a closure study between the optical aerosol particle properties in ambient state based on the airborne

ACTOS measurements and derived with the lidar measurements. The investigation of the optical properties shows for 14 measurement-points that the airborne-based particle light backscatter coefficient is for 1064 nm 50 % smaller than the measurements of the lidar system, 27.6 % smaller for 532 nm and 29.9 % smaller for 355 nm. These results are quite promising, since in-situ measurement based Mie-calculations of the particle light backscattering are scarce and the modelling is quite challenging. In contradiction for the particle light extinction coefficient retrieved from the airborne in situ measurements were found a good



agreement. The airborne-based particle light extinction coefficient was just 7.9 % larger for 532 nm and 3.5 % smaller for 355 nm, for an assumed lidar ratio (*LR*) of 55 sr. The particle light extinction coefficient for 1064 nm was derived with a *LR* of 30 sr. For this wavelength, the airborne-based particle light extinction coefficient is 5.2 % smaller than the lidar-measurements. Also, the correlation for the particle light extinction coefficient in combination with Mie-based *LR*'s are in agreement for typical *LR*'s of

European background aerosol.

## 1 Introduction

Aerosol particles are a ubiquitous constituent of the Earth's atmosphere (Vaughan and Cracknell, 2013). Their sources are manifold, reaching from natural ones such as the oceans, deserts and the biosphere to anthropogenic ones such as biomass burning activity, transportation, agricultural and re-suspended dust or industrial pollution (Pöschl, 2005; Seinfeld and Pandis, 2006). Once aerosol

particles are formed from precursor gases or suspended in air, they can be carried over hundreds to thousands of kilometers before they are removed from the atmosphere by dry or wet deposition. The lifetime in the boundary layer counts from hours to approximately two weeks. (Seinfeld and Pandis, 2006). During their residence time in the atmosphere, aerosol particles have impacts on atmospheric chemistry, cloud formation and microphysics (change of cloud-albedo; Twomey et al., 1977) as well as on the radiation budget by changing cloud albedo and cloud lifetime (Twomey et al., 1977). Consequently, aerosol particles have both a

natural and an anthropogenic influence on weather and climate (IPCC, 2013). The direct climatic effect of aerosols is based on their radiative cooling or heating of the atmosphere due to scattering and absorption of solar radiation (Bohren and Huffman, 1983; Chauvigné et al., 2016; Seinfeld and Pandis, 2006). The total radiative forcing due to aerosol-radiation interaction is assessed to be -0.35 W m⁻² (IPCC, 2013). Also the type of aerosol is important in this consideration. For instance, inorganic salts such as sulfate or nitrate aerosols lead to an estimated negative radiative forcing of -0.4 W m⁻² and therefore have a cooling effect on the atmosphere.

The absorbing behavior of black carbon (BC) particles in contrast warms the atmosphere and leads to a positive radiative forcing of approximately +0.71 W m⁻² (90 % uncertainty bounds from +0.08 to +1.27 W m⁻²) (Bond et al., 2013). These estimates are subject to uncertainties of 50 to 100 %. A considerable fraction of this uncertainty arises from the highly uncertain knowledge of the vertical distribution of the aerosol particles in the atmosphere. Exemplarily, Zarzycki and Bond (2010) found that small changes of the vertical BC distribution at cloud interfaces lead to a change in global radiative forcing by 5 to 10 %. Samset et al. (2013) furthermore

stated that at least 20 % of the uncertainty in radiative forcing due to the BC is caused by the diversity of the modeled BC particle mass vertical distribution. For aerosol types which contain hydrophilic aerosol compounds such as inorganic salts, also the vertical profile of the relative humidity (*RH*) needs to be known to determine the actual particle hygroscopic properties, but also in order to account for changes in the scattering properties due to hygroscopic-growth effects (Pilinis et. al, 1995).

In particular, aerosol particle properties in the planetary boundary layer (PBL) require a thorough characterization, because the

majority of the global aerosol mass is emitted, formed (Rosati et al., 2016b) and also trapped there (Summa et al., 2013). For instance, for a residential area in the Czech Republic, Hovorka et al. (2016) found aerosol particle mass concentrations at the top of the PBL being five times larger than just above the PBL (50 µg m⁻³ in contrast to 10 µg m⁻³).

In order to derive the aerosol radiative forcing in an atmospheric air column, profiles of the aerosol particle light extinction coefficient ($\sigma_{ext}$), which is the sum of the aerosol particle light absorption and scattering coefficient, are a feasible measure. Height-

resolved aerosol particle light extinction coefficients can be obtained either by airborne in situ measurements or with remote sensing



techniques such as lidar. Ground-based remote sensing observations with lidar are suitable to derive long-term temporally resolved profiles of the mentioned coefficients detecting the backscattered light of the total aerosol particle population in its ambient state (Baars et al., 2016; Engelmann et al., 2016). However, lidar measurements are restricted to the retrieval of the total aerosol particle light extinction coefficient. The separation into the contributions of scattering and absorption relies on complex inversion schemes,

which are restricted to nighttime observations, long averaging times, and rather low vertical resolution (Müller et al., 1999 and 2000). Recently, novel approaches based on the combination of daytime lidar observations with Sun-photometer measurements of column-integrated aerosol particle light scattering properties were developed, which allow estimating the contributions of absorption and scattering. This is for instance the case for the Generalized Aerosol Retrieval from Radiometer and Lidar Combined data (GARRLiC) algorithm (Lopatin et al., 2013). However, these techniques are in general still based on column-integrated

measurements and thus are still subject to considerable uncertainties, if the aerosol load is low (Bond et al., 2013). Furthermore, these methods are limited to certain conditions, such as the requirements of cloud free conditions and high aerosol optical depths of at least 0.5 at a wavelength of 440 nm (Dubovik et al., 2002).

A benefit of airborne and ground-based in situ measurements is that they allow to obtain high-quality measurements of the aerosol particle number size distribution (PNSD) and optical properties of the aerosol, and consequently the relationship between aerosol

microphysical properties, chemical properties and resulting aerosol particle light absorption, scattering and extinction coefficients. Especially, on ground a large number of long-term observations exists. For instance, in Germany the German Ultra-fine Aerosol Network (GUAN; Birmili et al., 2016) is operative. The Global Atmosphere Watch (GAW) network includes a large number of operating stations. Disadvantageously, with ground-based in situ measurements no vertically resolved information about aerosol properties are available, which are needed to ascertain aerosol-cloud interaction (Bréon, 2006). Without vertically resolved

informations, ground-based observations are usually assumed to be representative for the entire PBL and even ground-based measurements are often extrapolated to larger scales (Väänänen et al., 2016). Thus, as stated, e.g. by Rosati et al. (2016a), it is of scientific interest to better understand whether ground-based in situ measurements can be used to investigate aerosol properties, in particular their optical properties, for elevated atmospheric layers. This general approach leads to biases in modeling aerosol radiative effects. Especially, indirect effects indicated by anthropogenic emitted aerosol particles acting as cloud condensation nuclei

(CCN) are contributing strongest to the uncertainty in aerosol total radiative forcing (Bernstein et al., 2008; Schwartz et al., 2010). Recently, Mamouri and Ansmann (2016) provided a method to derive CCN number concentration (CCN-NC) profiles from lidar measurements. This method is the first step to evaluate CCN-NC's profiles with ground-based techniques. However, this method underlies significant uncertainties.

Opposed to the ground-based in situ measurements, airborne measurements, such as from aircraft (Wex et al., 2002), tethered-

balloon systems (Ferrero et al., 2014; Mazzola et al., 2016; Ran et al., 2016), zeppelin-systems (Rosati et al., 2016a and 2016b), unmanned-aerial-systems (UAS; Altstädter et al., 2015), or helicopter-borne payload (Siebert et al., 2006) are capable to provide spatio-temporal highly resolved measurements of optical and microphysical aerosol particle properties in a vertical and horizontal manner. However, these observations are rather expensive in cost and limited in time.

Both, airborne and ground-based in situ measurements alter the humidity state of the aerosol (Tsekeri et al., 2017). The aerosol is

often dried before the particle properties are characterized. Accounting for the hygroscopic properties of the aerosol particles, which can be measured or calculated, allows to simulate the ambient state aerosol optical and microphysical properties. The





parameterization by Petters et al. (2007) is a useful approach to ascertain the hygroscopic growth of the aerosol particles on the basis of their hygroscopicity parameter ($\kappa$).

Within the scope of this article, two of the above-mentioned challenges are addressed by means of a sophisticated closure studies: a) ground-based in situ observations to airborne in situ observations, and b) airborne in situ observations to ground-based remote

sensing. These were corroborated in the frame of the HD(CP)[2] Observational Prototype Experiment HOPE (Macke et al., 2017) at the Central European research observatory Melpitz, Germany. In particular, lidar-based aerosol optical properties are compared to respective values obtained from airborne in situ measurements, based on modeled optical properties for the regional background aerosol under consideration of the hygroscopic growth of the aerosol particles. We focus on the aerosol particle light backscatter coefficient ($\sigma_{bsc}$), since this is the directly measured property of a lidar system. Its conversion with the extinction-to-backscatter ratio

(lidar ratio, $LR$) to the particle light extinction coefficient is also subject of this investigation.

Second, the representativeness of ground-based observations of CCN-NC and thereby directly connected the aerosol hygroscopicity, particle number concentration (PNC), and the PNSD for different conditions in the PBL are studied by comparing the airborne in situ measurements with the observations at Melpitz. Furthermore, CCN-NC profiles derived with the approach of Mamouri and Ansmann (2016) are compared with in situ measured CCN-NC's for 0.2 % supersaturation.

**2 Experiment**

"HD(CP)² Observational Prototype Experiment"-Melpitz (HOPE-Melpitz) was one of two field experiments within the scope of the "High Definition Clouds and Precipitation for advancing climate prediction" project (see http://www.hdcp2.eu). The projects aims have been to reduce uncertainties in the representation of cloud and precipitation in atmospheric models (detailed information of HOPE is given in Macke et al., 2017).

HOPE-Melpitz took place between Sept. 9 and 27, 2013, at the Central European research observatory Melpitz, Saxony, Germany (51° 32' N, 12° 56' E; 84 m a.s.l). Melpitz, is located in a rural area, 44 km northeast of Leipzig. The approximate distances to the Baltic Sea in the north are 400 km, to the North Sea 500 km, and to the Atlantic Ocean 1000 km, respectively. The TROPOS field observatory Melpitz is situated in a plain open landscape, bounded by the Ore Mountains to the further south, Berlin to the north, Leipzig to the west and Polish industrial areas to the east. The measurements are therefore representative for the Central European

regional background aerosol.

The Melpitz Observatory is included in several observational networks and set-ups, such as LACROS (Leipzig Aerosol and Cloud Remote Observations System), GUAN (German Ultra-fine Aerosol Network; Birmili et al., 2016), ACTRIS (Aerosols, Clouds and Trace gases Research Infrastructure; www.actris.eu), and GAW (Global Atmosphere Watch; http://www.wmo.int/pages/prog/arep/gaw/aerosol.html). A ground stock of instruments is implemented for permanent, high-quality

long-term measurements, including PNSD, CCN-NC, aerosol particle light scattering and absorption, as well as aerosol chemical composition. A detailed description of this measurement site is given in Spindler et al. (2013 and 2010).

In addition to the continuously operating instrumentation, several ground-based remote-sensing instruments (e.g. the Raman-lidar system Polly[XT]; Engelmann et al., 2015) were installed during the intensive campaign period providing a detailed overview of the atmospheres constitution (cf. Figure 1). These measurements were complimented by the helicopter-borne payload ACTOS (Airborne





Cloud and Turbulence Observation System; Siebert et al, 2006) inferring microphysical aerosol particle and cloud properties with a high spatio-temporal resolution. Figure 1 shows a scheme of the installed instrumentation during the HOPE-Melpitz campaign. The following section will provide a detailed description of the instrumentation used within the scope of this work.

### 2.1 Ground-based in situ instrumentation

### 2.1.1 Particle number size distribution

The PNSD was derived using two instruments under controlled dry conditions as recommended in Wiedensohler et al. (2012). A dual mobility particle size spectrometer (TROPOS-type TSMPS; Birmili et al., 1999) was used to measure the PNSD in the mobility diameter $D_{em}$ range from 3 to 800 nm. Each scan of the PNSD lasts 10 minutes and is available every 20 minutes. An aerodynamic particle size spectrometer (model TSI APS-3320, Inc., Shoreview, MN USA) was employed to determine the PNSD in aerodynamic

diameter $D_a$ range from 0.8 to 10 µm, also with a time resolution of 10 minutes. The TSMPS PNSD was derived using the inversion algorithm of Pfeifer et al. (2014) and corrected with respect to internal and inlet diffusional losses, using the method of "equivalent pipe length" (Wiedensohler et al., 2012).

Both size-distributions where merged to a continuous distribution after converting the $D_a$ of the APS to $D_{em}$ by using:

$$D_{em} = D_a \sqrt{\frac{\rho_0}{\rho_a}}, \qquad (1)$$

according to DeCarlo et al. (2004), whereby the aerosol particle density is assigned by $\rho_a$ and $\rho_0$ is the standard density of 1.0 g cm$^{-3}$. In this study we assumed an effective aerosol particle density of 1.6 g cm$^{-3}$, according to Ma et al. (2014) for the fine-mode aerosol. The effective density combines the particle density and shape.

### 2.1.2 Chemical composition

This section introduces instruments used for measuring the aerosol particle composition, including water-soluble non-refractory

particulate matter and water-insoluble black carbon (BC).

**Water-soluble compounds**

In this study, a data-set of the continuously running Quadrupole Aerosol Chemical Speciation Monitor (Q-ACSM, Aerodyne Res. Inc, ARI, Billerica, MA.; Ng et al., 2011) was used. This instrument measured the mass concentration of non-refractory particulate matter in the fine regime (NR-PM1) with a resolution of roughly 25 minutes. Mainly, the Q-ACSM measures $SO_4^{-2}$, $NO^{-3}$, $NH^{+4}$

and organics (Gysel et al., 2007). A detailed description of the instrument is provided in Ng et al. (2011) and Fröhlich et al. (2015). Based on these ion measurements, the chemical composition of the aerosol particles itself was derived by a simple ion pairing scheme published by Gysel et al. (2007). The species of the aerosol compounds derived with the Q-ACSM we considered as water-soluble. The major mass fraction of water-soluble compound are in $PM_1$ and are thus also representative for $PM_{2.5}$.

**Equivalent Black carbon (eBC)**

The Multi Angle Absorption Photometer (MAAP; Model 5012, Thermo Scientific) was used to derive the equivalent mass concentration of the non-water-soluble black carbon (eBC) for $PM_{10}$ aerosol. A MAAP determines the aerosol particle light





absorption coefficient ($\sigma_{abs}$) by measuring the attenuation of light at a wavelength of 637 nm (Müller et al., 2011) due to particulate matter deposited on a filter band and by reflected light at two angles. The eBC particle mass concentration is calculated by a mass absorption cross section of 6.6 m² g⁻¹. With the assumption that all of the measured eBC is elemental carbon (EC), according to Spindler et al. (2013) and Poulain et al. (2014) we assume here that PM$_1$ aerosol contains 90 % of the PM$_{10}$ eBC (EC) mass derived

with the MAAP.

The particle volume concentration and as a consequence thereof the volume fraction of each aerosol particle compound was calculated by using the density of the individual species (see Table 1). Like Tsekeri et al. (2017), we assumed that the aerosol particles in PM$_{2.5}$ and PM$_1$ had the similar chemical composition since no highly time-resolved chemical composition measurements for coarse-mode aerosol particles were available during the campaign.

**2.1.3 Cloud Condensation Nuclei number concentration**

Ground-based monodisperse CCN-NC measurements at Melpitz are part of the standard measurements within the ACTRIS network. A stream-wise thermal gradient cloud condensation nuclei counter (CCNc; mod. CCN-100, Droplet Measurement Technologies, Boulder, USA; Roberts and Nenes, 2005) is operated to investigate the supersaturation dependent growth-activation of particles. The relative uncertainty of the supersaturation can be estimated to be within 10% (Henning et al., 2014).

Briefly, the measurement method is as follows; a differential mobility analyzer (DMA) selects aerosol particles according to their mobility diameter, which are then counted in total number at this size with a particle counter (model TSI CPC-3010, Inc., Shoreview, MN USA $N_{tot}(D_p)$) and at a certain water-supersaturation with the CCNc ($N_{CCN}(D_p)$). The size-dependent activated fraction (*AF*) was calculated by the ratio of the PNC of activated particles and the total PNC of a certain size measured after the DMA. The *AF* was derived on the basis of diameter scans in the size-range from 20 to 440 nm (dry diameter of the aerosol particles) and for

different supersaturations in the range from 0.1 % to 0.7 %. With a Gaussian error-function the *AF* can be fitted according to:

$$AF = \frac{a+b}{2}\left[1 + erf\left(\frac{D-D_c}{\sigma\sqrt{2}}\right)\right], \tag{2}$$

where *a* and *b* denote the upper and the lower limit for the calculation of the critical diameter $D_c$ (Henning et al., 2014). $D_c$ is the diameter from which on 50 % of the particles are activated to droplets. With the single-parameter parametrization by Petters and Kreidenweis (2007) and $D_c$ from Eq. (2) the hygroscopicity parameter can be derived by using:

$$\kappa = \frac{4A^3}{27D_c^3 (\ln SS)^2}, \tag{3}$$

with

$$A = \frac{4\sigma_{s/a}M_W}{RT\rho_W}. \tag{4}$$

In Eq. (3) and (4), $\rho_W$ is the density of water, $M_W$ the molecular weight of the water, *SS* the supersaturation inside the CCNc, $\sigma_{s/a} = 0.072$ J m⁻² the surface-tension of the solution, $R = 8.314$ J mol⁻¹ K⁻¹ the universal gas constant and *T* the temperature.





### 2.2 Airborne measurements

The Airborne Cloud and Turbulence Observation System, ACTOS (Siebert et al., 2006), was deployed at a 140 m long rope below a helicopter (Siebert et al., 2006). Airborne in situ measurements were performed on seven days between Sept. 12 and 28, 2013. Each flight lasted typically between 90 and 120 min (cf. Table 2). The measurement flights started at the small airport of Beilrode
approximately 11 km ton the northeast of Melpitz (see Figure 2). The flights were usually performed as follows: after the arrival in the measurement area of Melpitz, a vertical profile up to an altitude of 2300 m above ground was performed first to determine the layer structure of the atmosphere. In a second step, legs of up to 20 minutes with constant heights were carried out. In this study, these parts are signed as horizontal legs.

ACTOS includes instruments to provide meteorological parameters, including relative humidity $RH$ and temperature $T$ with a time
resolution of 100 Hz. ACTOS probes the atmosphere with a true air speed of around 20 m s$^{-1}$. Real-time data allow the on-board scientist to observe actual atmospheric conditions and to adjust the flight pattern accordingly.

In addition to the meteorological sensors, also the PNC and PNSD were determined on ACTOS (Wehner et al., 2010; Wehner et al., 2015; Ditas et al., 2012). According to recommendations given in Wiedensohler et al. (2012), the aerosol flow was dried, using a silca-based diffusion dryer to obtain a $RH$ below 40 %. A mobility and an optical particle size spectrometer (MPSS and OPSS) were
employed to determine the PNSD in the size-range of 8 nm to 2.8 µm. In the further course of this work, PNSD connotes dry state PNSD.

A TROPOS-type mobility particle size spectrometer (MPSS) measured the PNSD in the size range from 8 to 226 nm (mobility diameter $D_{em}$) with a time resolution of 120 s. A Grimm optical particle size spectrometer (OPSS; model Grimm 1.129 (skyOPC); Grimm Aerosol Technik, Ainring, Germany) was used to obtain the PNSD in the size range from 356 nm to 2.8 µm (optical diameter
$D_o$) with time resolution of one second. A full PNSD was derived by combining each of the SMPS-PNSD with the respective 120-s median OPSS-PNSD. This set-up causes uncertainties in integration based aerosol properties, such as the total aerosol particle number concentration, because integrals of the non-observed size range were approximated with a trapezoid.

The MPSS consists of a) a bipolar diffusion charger to bring the aerosol particle population into the bipolar charge equilibrium (Fuchs, 1963; Wiedensohler, 1988), b) a TROPOS-type differential mobility analyzer, DMA (Hauke-type, short) to select the aerosol
particles with respect to their electrical mobility, c) and a condensation particle counter (CPC, Model 3762A, TSI Inc., Shoreview, MN USA) with a lower detection efficiency diameter of 8 nm. This set-up was also used in Wehner et al. (2010) and Ditas et al. (2012). The measured raw PNSD of the MPSS was processed using the inversion algorithm of Pfeifer et al. (2014) by enhancing the inversion with the PNSD obtained with the OPSS. The PNSD was also corrected with respect to the sampling efficiency of the inlet according to Kulkarni et al. (2011). With a sampling angle $\alpha_s = 85°$ and a volume flow of 3.7 liter per minute the inlet had a
theoretical upper 50 % cut-off aerodynamic diameter of approximately $D_{p,50} = 2$ µm. Furthermore, the measured PNC of ultrafine particles is influenced by diffusional losses. Following Kulkarni et al. (2011) and Wiedensohler et al. (2012) these losses were corrected using the method of the "equivalent pipe length". A second CPC, identically to the CPC consisting in the MPSS, was installed to determine PNC ($N_{CPC}$) of the aerosol sampled through the same inlet of the MPSS with a temporal resolution of 1 Hz and a lower cut-off of ~8 nm. This second CPC allowed furthermore to evaluate the quality of the PNSD measurements.
Since the Grimm OPSS was not calibrated with spherical PSL particle size standards, it was not possible to adjust the optical PNSD with a refractive index typical for the atmospheric aerosol in Germany. Therefore, the here used OPSS measurements deviate from the "real" PNSD to some extent.



Furthermore, the polydisperse CCN-NC was determined with a mini cloud condensation nuclei counter (mCCNc, custom built by G. C. Roberts) also installed on ACTOS. The CCN-NC derived with the mCCNc ($N_{\text{CCN,mCCNc}}$) was measured at a supersaturation of 0.2 % (within an accuracy of 10 %; Henning et al., 2014).

**2.3 Ground-based remote sensing**

A 3+2 wavelength (3 channels for backscatter and 2 channels for extinction) polarization lidar system, called Polly$^{\text{XT}}$ introduced by Engelmann et al. (2016), was used to evaluate vertical profiles of optical aerosol properties. In particular, the particle backscatter coefficient $\sigma_{\text{bsc}}$ was derived for 355, 532 and 1064 nm. Furthermore, Polly$^{\text{XT}}$ is capable to derive the $\sigma_{\text{ext}}$ for 355 and 532 nm. In this paper, aerosol particle optical properties derived with the lidar system are assigned with the subscript "lid".

Briefly, the here used lidar system contains a Nd:YAG laser, which emits laser pulses at 20 Hz. The full overlap of the laser beam
and the receiver field of view (FOV) for this system is at about 800 m height. Below this height, an overlap correction can be applied. The experimental determination of the overlap-height is described in (Wandinger and Ansmann, 2002). Measurements of the lidar system were available each 30 s with a vertical resolution of 7.5 m.

As the signal-to-noise ratio in the channels of the Raman scattered light is too weak during day time, no independent particle light extinction profiles are available. Therefore, the extinction-to-backscatter ratio, or lidar ratio ($LR$ in sr), an aerosol type depending
intensive property, was used to convert $\sigma_{\text{bsc}}$ to $\sigma_{\text{ext}}$ by:

$$\sigma_{\text{ext}} = LR \times \sigma_{\text{bsc}}. \tag{6}$$

Several studies (e.g. Tao et al., 2008; Lu et al., 2011; Ferrare et al., 2001; Müller et al., 2007; Haarig et al., 2016) investigated the $LR$ for different atmospheric conditions and aerosol types, like dust in Groß et al. (2011) and volcanic ash in Ansmann et al. (2010). The studies showed that the $LR$ is a highly variable parameter depending on the predominant aerosol. In this study we used a height-
constant $LR$ of 55 sr to derive profiles of $\sigma_{\text{ext}}$ for 355 and 532 nm on the basis of the lidar measurements, which is representative for the aerosol types expected in Melpitz and in agreement with values observed during night time at the measurement location. A height independent $LR$ of 30 sr for 1064 nm provided by Omar et al. (2009) was used in this study. This assumption might introduce errors in the retrieval of $\sigma_{\text{ext}}$. Overall, we consider an uncertainty in the lidar measurements of up to 15 %.

We use here a fixed $LR$, which is in agreement with the Raman-measurements (direct measurement of $LR$; Ansmann et al., 1992)
during night at the respective period. These fixed $LR$ also fit to long-term observations of different aerosol types at other European continental sides and aerosol types (clean and polluted continental aerosol, mineral desert dust, and smoke, Baars et al., 2016; Groß et al., 2013; Mattis et al., 2004; Müller et al., 2007; Schwarz, 2016).

Integrating the derived profiles of $\sigma_{\text{ext,lid}}$ yields the aerosol optical thickness ($AOD$), which was compared with on-going AERONET (Aerosol Robotic Network; http://aeronet.gsfc.nasa.gov/; station: Melpitz) Sun-photometer measurements at wavelengths of 340,
500, and 1020 nm. Both measurements agree well within the uncertainties, which were relatively high due to the very low $AOD$ (e.g. on Sept. 14: $0.014 \pm 0.001$ for 1020 nm, $0.087 \pm 0.004$ for 500 nm and $0.158 \pm 0.004$ for 340 nm between 11.50 and 12.20 UTC).

Besides the validation of the $LR$ for the three wavelengths, we also considered a new method provided by Mamouri and Ansmann (2016) to derive CCN-NC profiles from lidar measurements ($N_{\text{CCN,lid}}$). This method converts particle light extinction coefficients to
number concentration of CCN for different supersaturations and different aerosol types. For continental aerosol (subscript "c"):





$$n_{\mathrm{CCN,ss,c}}(z) = f_{\mathrm{ss,c}} \times n_{50,\mathrm{c,dry}}(z), \tag{7}$$

with:

$$n_{50,\mathrm{c,dry}}(z) = c_{60,\mathrm{c}} \times \sigma_{\mathrm{ext}}^{x_{\mathrm{c}}}(z), \tag{8}$$

has to be applied (Mamouri and Ansmann, 2016) in accordance to Shinozuka et al. (2015). Here, $n_{\mathrm{CCN,ss,c}}$ assigns the CCN-NC at

given supersaturation $ss$ and height $z$ in cm$^{-3}$. The PNC of particles with a diameter larger than 100 nm is symbolized by $n_{50,\mathrm{c,dry}}$ (50 nm radius). $c_{60,\mathrm{c}}$ assigns the conversion factor in cm$^{-3}$ for the ambient aerosol particle light extinction coefficient ($\sigma_{\mathrm{ext}}$) in Mm$^{-1}$. And $x_{\mathrm{c}}$ is the aerosol extinction exponent.

For the here presented cases Mamouri and Ansmann (2016) provided a value of 1.0 for $f_{\mathrm{ss,c}}$ for a supersaturation of 0.15 %. Therefore, retrieved concentrations of CCN may underestimate direct measured CCN concentrations of the mCCNC on ACTOS. Furthermore,

for $x_{\mathrm{c}}$ they estimated $0.94 \pm 0.03$ for Germany and a lidar wavelength of 532 nm. Also they provided for $c_{60,\mathrm{c}}$ a value of $25.3 \pm 3.3$. $n_{50,\mathrm{c,dry}}$ and consequently $n_{\mathrm{CCN,ss,c}}$ can be retrieved with an uncertainty of factor 2 (Mamouri and Ansmann, 2016).

## 3. Methodology

In this chapter we will provide an overview of the data set used in this investigation and the model that is used to determine the aerosol particle optical properties.

### 3.1. Case studies

From the eight ACTOS flights of the intensive measuring period (see Table 2), three were taken due to preferred conditions and thus will be intensively discussed (flights: 20130914a, 20130914b, 20130927a, in the following abbreviated as 14a, 14b, 27a). The major preferential condition was clear skies in all altitudes levels in order to prevent the influence of the clouds on *AOD* measurements of the Sun-photometer and to ensure that the lidar covers the entire atmospheric column.

Figure 3 and Figure 4 show the time-elevation plot of the range-corrected attenuated backscatter signal of the lidar system. White areas in the figures represent high backscattering, mostly by clouds. Blue or black areas represent low light backscattering and thus regions of very clean air. Red and yellowish colors indicate enhanced light backscattering by aerosol particles. The overlaying solid black line indicates the height of ACTOS during a measurement flight. Capital letters mark horizontal parts of the flight investigated later in this study.

In particular, during the flights on Sept. 14, 2013, episodes of a cloud free air column above the lidar were apparent (before leg D on flight 14a, between 10.15 and 12.30 UTC, during leg D on flight 14b). Cloud free periods did occur during flight 27a (clouds visible around 10.35, and from 10.50 to 11.30). Furthermore, for Sept. 14, 2013, a residual layer is visible between 8 and 10 UTC reaching a height of up to 1800 m. Its thickness decreased during daytime and the residual layer vanished at around 12.00 UTC. At the same time, a well pronounced mixing layer was built-up. Its upper boundary is characterized by a sharp gradient of the backscatter

signal (Figure 3). The development of the mixing layer is visible in the lidar measurements from 9.00 UTC and it reached a height of about 1600 m at 14.00 UTC.

During the measurement flights, Melpitz was dominated by marine air masses influenced by continental pollution. Exemplary, for Sept. 14 and Sept. 27, 2013, three 72 hours backward-trajectories for the height of 500 (red lines), 1000 (blue lines) and 1500 m



(green lines) above ground are shown in Figure 5 and Figure 6. These trajectories where calculated using the Hybrid Single-Particle Lagrangian Integrated Trajectory (HYSPLIT) model of the air resource laboratory (ARL) of the National Oceanic and Atmospheric Administration (NOAA). HYSPLIT is available at http://www.ready.noaa.gov. A detailed description of the model is available in Stein et al. (2015).

For Sept. 14, 2013, a westerly flow in all heights was apparent. The air masses crossed the North Sea before traveling across the continent to Melpitz. Furthermore, for Sept. 27, 2013, the air masses were subsiding during the last 36 h crossing the Baltic Sea. The three air parcels reaching Melpitz in 500, 1000 and 1500 m originated from Scandinavia and proceeded southwards. In contrast to Sept. 14, 2013, the air parcel with the lowest height in the beginning (green line) in roughly 1500 m above ground and reached Melpitz at a higher altitude (1500 m) than the air parcel marked by the red and blue, originating from a height of roughly 3000 m.

**3.2 Airborne in situ aerosol optical properties**

In this study, the calculation of aerosol optical properties was performed on the basis of Bohren and Huffman (1983). The complex refractive index, the hygroscopicity and the mixing of the aerosol particles is needed to compare calculated optical properties with measured ones. A scheme of our method is shown in Figure 7. The method and its application is described in the following.

The mixing state can be assumed by different mixing approaches. The dry state optical closure study by Ma et al. (2014) 
shows that the approach of internally mixed coated (aerosol particles consists of a core surrounded by a shell; core-shell approach (CS)) aerosol particles results in the best agreement between modeled and measured hemispheric backscatter coefficients for Melpitz. Furthermore, Zhang and Thompson (2014) and Kahnert et al. (2012) discussed the mixing morphology and its influence on particle light absorption and scattering. Zhang and Thompson found, that the core-shell mixing assumption lead to higher modelled particle absorption than the approach of internally homogeneous mixed particles (24 % difference, 115 % in maximum), 
especially when the core of light absorbing carbon is small compared to the shell. In contrast, for particle light scattering they did not observe a significant difference between both approaches. Kahnert et al. (2012) showed that the core-shell model underestimates the particle light absorption but reproduces the particle light extinction sufficient. In conclusion, the mixing approach used in this study is applicable for modelling aerosol particle light extinction.

This discussion in the previous paragraph implies, although the particle light absorption is overestimated, that the core-shell mixing 
assumption is satisfying for the aerosol apparent in Melpitz. That means that in this work it is assumed that the aerosol particles consist of a core of water-insoluble highly-absorbing soot (eBC) and a shell of water-soluble less-absorbing material, which includes organic matter, ammonia nitrate, and sulfate species.

The Mie-code calculates the scatter-, extinction-, absorption- and backscatter-efficiency of a single, spherically symmetric aerosol particle with a given complex refractive index of its shell and core and a given diameter of the core and thickness of the shell.

The goal of this study is to investigate the aerosol particle light extinction ($\sigma_{\mathrm{ext}}$) and backscatter coefficient ($\sigma_{\mathrm{bsc}}$) in ambient state. $\sigma_{\mathrm{bsc}}$ can be calculated with Eq. (7) and $\sigma_{\mathrm{ext}}$ with Eq. (8) (adapted and modified from Virkkula et al., 2011):

$$\sigma_{\mathrm{bsc}}(\lambda) = \frac{1}{4\pi} \int Q_{\mathrm{bsc}}(\lambda, D_{\mathrm{p}}, n) \frac{\pi D_{\mathrm{p}}^2}{4} \frac{\mathrm{d}N(D_{\mathrm{p}})}{\mathrm{d}\log D_{\mathrm{p}}} \mathrm{d}D_{\mathrm{p}}, \tag{7}$$

$$\sigma_{\mathrm{ext}}(\lambda) = \int Q_{\mathrm{ext}}(\lambda, D_{\mathrm{p}}, n) \frac{\pi D_{\mathrm{p}}^2}{4} \frac{\mathrm{d}N(D_{\mathrm{p}})}{\mathrm{d}\log D_{\mathrm{p}}} \mathrm{d}D_{\mathrm{p}} \tag{8}$$



Hereby $Q_{\text{bsc}}(\lambda, D_p, n)$ and $Q_{\text{ext}}(\lambda, D_p, n)$ are the backscatter and extinction efficiency, respectively, of aerosol particles with a diameter $D_p$ and a complex refractive index $n$ at a wavelength $\lambda$ (Virkkula et al., 2011). The equations to derive the particle light scattering efficiency for coated particles is provided by Bohren and Huffman (1983); Dombrovsky (2011). $\frac{dN(D_p)}{d \log D_p}$ denotes the PNSD of the aerosol and can be measured by particle size-spectrometers (see Sect. 2.2 and Sect. 2.3.2).

Aerosol particles in humid ambient conditions are underlying a growth due to water vapor uptake. The magnitude of growth depends on particle size, hygroscopicity parameter $\kappa$, and ambient $RH$. Hygroscopic growth changes size, shape, and the complex refractive index of aerosol particles. The change of shape is not considered in this study because the particles are assumed to be spherical in dry-state anyway.

Measurements of the aerosol particle chemical composition (see Sect. 2.2.2) provided volume fractions of aerosol particle
compounds such as organic and black carbon, ammonium nitrate, ammonium sulfate. A volume weighted sum of $\kappa$ of the aerosol particle compounds provided a mean $\kappa$ of the entire aerosol particle. Measurements of cloud condensation nuclei may also provide $\kappa$ as described in Henning et al. (2014) and Sect. 2.2.3 (cf. Figure 7, rhombus with question marks).

Here, we assumed that in the dry state each aerosol particle consists of the same constant volume fraction of each component, because no size-resolved particle chemical composition measurements with a high time resolution were available. Petters and
Kreidenweis (2007) provided a semi-empirical parametrization for the diameter of a particle with a given hygroscopicity in ambient conditions as a function of $RH$ and $T$. Using this parametrization allows to derive the PNSD in ambient state.

The difference in the volume of the aerosol particles in ambient and dry state is the total volume of the water $V_{\text{wat}} = V_{\text{aer,ambient}} - V_{\text{aer,dry}}$ accumulated on the aerosol particles (green rectangle in the scheme). A detailed description of the Köhler-theory is given in Köhler (1936).

The complex refractive index of the particle core (eBC) is known. In dry state, the shell consists of different water-soluble compounds (subscript "s"), which are assumed to have the same complex refractive index (see Table 3). In ambient state the aerosol particle shell is made up of the water-soluble material and the water (subscript "w") itself. The algorithm used here applied a volume-weighted sum to derive the complex refractive index $n = n_{\text{re}} - i n_{\text{im}}$ of the aerosol particle shell in the ambient state:

$$n_{\text{shell,amb}} = f_{v,s}(n_{re,s} + i\, n_{im,s}) + f_{v,w}(n_{re,w} + i\, n_{im,w}), \tag{9}$$

where $f_{v,s}$ is the volume fraction of the water-soluble compounds in the shell and $f_{v,w}$ the volume fraction of the water. $n_{re,s}$, $n_{re,w}$ are the real part of the complex refractive index of the water-soluble material and the water respectively, and $n_{im,s}$ and $n_{im,s}$ denote the imaginary part of the refractive index of the soluble material and the water, respectively. The complex refractive index of water and the water-soluble material are shown in Table 3.

Furthermore, the diameter of the light absorbing eBC core for each aerosol particle has to be taken into account for the
calculation of the aerosol optical properties. With the volume fraction of the eBC ($f_{v,\text{eBC}}$) derived from the chemical composition measurements (see Sect. 2.2.2), the diameter of the eBC core ($D_{\text{eBC}}$) of each aerosol particle with a diameter of $D_p$ can be calculated according to Ma et al. (2014):

$$D_{\text{eBC}}(D_p, f_{v,\text{eBC}}) = D_p f_{v,\text{eBC}}^{1/3}. \tag{10}$$

A Monte-Carlo simulation, also used in (Ma et al., 2014), is implemented in the here provided method to cover a possible range of
results of $\sigma_{ext}$ and $\sigma_{bsc}$ introduced by measurement uncertainties in the input parameters and due to their spatio-temporal variability.



A calculation of $\sigma_{ext}$ and $\sigma_{bsc}$ grounds on a PNSD of the aerosol. The Monte-Carlo simulation repeats the calculation of $\sigma_{ext}$ and $\sigma_{bsc}$ 50 times varying the input parameters within the respective uncertainties and standard deviations of mean uniformly distributed.

For the considered period (e.g. length of a horizontal leg) the mean and the respective standard deviation was calculated from the measurements of the PNSD and the prevalent ambient $RH$ and $T$. The aerosol hygroscopicity was derived by averaging the volume

fractions of each considered species on the basis of the Q-ACSM and MAAP measurements determined between 8.00 and 15.30 UTC, which cover the range of the flight times. The complex refractive index of the aerosol particles was calculated according to the mixing-rule introduced by Eq. (9), where each complex refractive index of the considered aerosol component (water, water-soluble and insoluble) was varied within its uncertainties given in Table 3. By calculating the average of the output of the 50 simulations the algorithm provides the average optical properties for the aerosol particles in their ambient state as well as the

uncertainty range due to the variability of the input parameters. A larger number of simulations does not change the standard deviation of the output.

In the following, the aerosol optical properties calculated with this algorithm on the basis of in situ measurements are assigned with the subscript "mie".

### 4 Results

In this section ground-based measurements will be related to vertical profiles to investigate the representativeness of in situ measurements on ground for the PBL. Furthermore, the results of the in situ based calculations and measurements from the lidar will be shown and compared.

#### 4.1 Representativeness of ground in situ measurements for the PBL

In this section we compare the PNSD, the aerosol particle number concentration (PNC) and the concentration of the cloud

condensation nuclei (CCN-NC) measured on ground and above ground with ACTOS.

#### 4.1.1 Particle number size distribution

The scans of PNSD in leg A, D, E and F of flight 14b were performed within the mixing layer (cf. Figure 3), while the scans of leg B and C were done above the mixing layer. A comparison of PNSD's measured during leg C and B with ground-based measurements is thus not useful. We focus therefore on the PNSD of leg A, D, E, and F.

The averaged PNSD's at standard conditions of leg A, D, E, and F are shown in Figure 8 (solid lines). The corresponding PNSD measured at Melpitz observatory is shown as dashed line with the respective color in each figure. Since there was no scan of ground-based PNSD available during leg D and E, the average of the PNSD one scan before and after the respective leg were taken for these legs. For the selected case, the ground-based PNSD agrees with the PNSD of leg A, E, and F in the size range of 30 to 100 nm within 10 %. For aerosol particles smaller than 30 nm, the difference between the curves increases, however the shape of both number size

distributions is similar. In the size range of the accumulation mode (100 to 500 nm) the mean airborne PNSD of leg E and F were up to two times larger than the PNSD observed on ground at the same time. This clearly corresponds with the integrated aerosol PNC recorded with the OPSS ($N_{OPSS}$) on ACTOS (cf. Figure 9), where the total PNC derived with the CPC ($N_{CPC}$) increases with height. During the first profile of flight 14b $N_{OPSS}$ increases with increasing height up to ~650 m (18 cm$^{-3}$ to ~45 cm$^{-3}$). The





measurements in leg D were performed at the top of the planetary boundary layer and therefore probably may have been influenced by mixing processes of clean air of the free troposphere and the more polluted air within the PBL. This explains the different shape and concentration of the PNSD of leg D in comparison to the ground-based measurements.

Differences in the airborne and ground-based PNSD may also occur due to horizontal inhomogeneity. Exemplary, Figure 9 shows the mean $N_{OPSS}$ measured within a layer between 950 and 1050 m height between 13.06 and 13.34 UTC on Sept. 14 (measurements during leg D and F). The red triangle symbolize the measurement site in Melpitz. The more reddish the color is the larger the PNC.

During leg D and F of flight 14b, the horizontal distance between ACTOS and Melpitz was between 500 m and 4700 m. Within this distance the aerosol PNC, PNSD and chemical composition may differ in the observed altitude. During the leg from south to north, the PNC varied by a factor of two, probably due to local influences on the transported pollution. This example demonstrates the horizontal variation of number concentration and potential deviation between ground based and vertical measurements due to the horizontal distance.

In conclusion, for aerosol particles larger than 30 nm we can state for the here presented case that ground-based measurements of the PNSD are representative for higher atmospheric layers within the PBL. For smaller particles, local events alter the PNSD and cannot be detected by ground-based measurements. The agreement is best for measurements of the PNSD in the lowest available altitude. Ground-based measurements are not representative for the observations near the top of the PBL. Here, entrainment and mixing processes affect the aerosol.

### 4.1.2 Aerosol particle total number and cloud condensation nuclei concentration

Figure 10a) and Figure 10b) each show two different profiles (black and blue) of the PNC measured with the CPC on ACTOS ($N_{CPC}$, left in each panel) and the CCN-NC ($N_{CCN}$, right in each panel) recorded with the mCCNC on ACTOS for flight 14b (panel a)) and 27a (panel b)). Additionally, the integrated PNC (left of each panel) of the PNSD and the CCN-NC (right of each panel) measured at the Melpitz observatory are shown (red crosses). Furthermore, CCN-NC profiles are shown derived on the basis of the approach of Mamouri and Ansmann (2016) (solid lines in left panels, shaded area marks the uncertainty). The first profile of flight 14a (black) was measured between 12.05 and 12.27 UTC. The second (blue) between 13.47 and 13.54 UTC. The respective measurements of the integrated PNC were sampled at 11.30, 11.50 and 12.10 UTC. For the second profile, the respective PNC at Melpitz observatory was measured at 13.10, 13.50 and 14.10 UTC, CCN-NC at 10.33, 12.43 and 14.53 UTC respectively. The first profile of flight 27a (black) was taken at the beginning of the measurement flight (between 10.24 and 10.34 UTC) whereas the second profile (blue) was conducted in the fully developed mixing layer between 11.29 and 11.36 UTC (see Figure 4; ascending part after leg D of the flight 27a).

The first profile of flight 14a (Figure 10a), black) shows an inversion at 1150 m altitude where $N_{CPC}$ decreased from 13000 to around 1000 cm$^{-3}$ (top of PBL). The layer below 350 m altitude (part of the flight from Beilrode to Melpitz) shows a two times smaller concentration than the layer above. Since the first part of the profile was performed on the way to Melpitz, a horizontal variability of the aerosol might be the reason for this behavior. The two distinct peaks (up to 12000 cm$^{-3}$) in the lower part of the $N_{CPC}$ profile are probably caused by exhaust-gases of the helicopter because an increased $CO_2$ concentration was measured at the same time. Above the lower part the atmosphere is well mixed between 350 and 1150 m altitude with a stable $N_{CPC}$ in the range from 10000 to 12000 cm$^{-3}$ and slightly larger PNC below the inversion. The $N_{CPC}$ recorded during the second profile of flight 14b (blue) increases





slightly with height. The second profile of flight 14b was completely located within the PBL since no sharp decrease of $N_{CPC}$ with height was observed.

Aerosol measurements at the observatory in Melpitz showed an event of high PNC between 12 and 14 UTC. The elevated PNC is probably caused by a transported plume since the $SO_2$ concentration increased by a factor of 10 at the same time (see Figure 11).

This advected plume was obviously not lifted into higher atmospheric layers. Thus, the ground-based measurements are decoupled from those in higher altitudes and are therefore not representative for the planetary boundary layer in this case.

In contrast to the ground-based measurements in Melpitz and excluding the case when the exhaust-gases influenced the airborne measurements, the measurements of the CPC at the surface in Beilrode were representative for the atmospheric layers above, since the PNC is as high as in higher atmospheric layers.

Airborne measurements of $N_{CCN}$ during the first profile of flight 14b (black dots) started above the top of the mixing layer and are therefore not of further interest. In the second profile (blue dots) $N_{CCN,mCCNc}$ varies between 886 and 2474 cm$^{-3}$ with an average of 1456 ± 301 cm$^{-3}$. The second profile was taken between two ground-based measurements (12.43 and 14.53 UTC). At both times the ground-based measurements in Melpitz resulted in smaller CCN-NC (600 and 976 cm$^{-3}$) than in in higher altitudes. In contrast, the lowermost measurements of the mCCNC (1279 ± 91 cm$^{-3}$ within 100 and 130 m altitude) on ACTOS (considered as measurement

on ground) in Beilrode do represent the measurements during the last profile of flight 14b. Spatial variability may explains that the ground-based CCN-NC measurements in Melpitz are not representative for collocated vertical profiles. In contrast, the lowermost CCN-NC measurements (~ 700 cm$^{-3}$) derived with the mCCNC on ACTOS are representative for the higher atmospheric layers.

In the first profile of flight 27a (black) the top of the mixing layer is around 250 m altitude marked by a sharp decrease in both $N_{CPC}$ and $N_{CCN,mCCNc}$. In the second profile (blue) three distinct layers are apparent. Up to a height of around 600 m $N_{CPC}$ and $N_{CCN,mCCNc}$

are almost constant at around 2000 cm$^{-3}$ and 600 cm$^{-3}$, respectively. Between 600 and 1050 m altitude an atmospheric layer was apparent with aerosol highly variable in $N_{CCN,mCCNc}$ and $N_{CPC}$. Compared to the layer below $N_{CPC}$ is up to six times and $N_{CCN,mCCNc}$ up to two times larger. Above that $N_{CPC}$ is constant at around 1000 cm$^{-3}$ with a sharp increase in the highest 50 meters of the profile. Note that $N_{CPC}$ in the highest layers shows the same values as in the first profile in this height. In the layer above 1050 m $N_{CCN,mCCNc}$ shows a slight decrease from around 500 to 100 cm$^{-3}$.

For both profiles, the ground-based measurements (Melpitz) of both the PNC and the CCN-NC agree with the airborne measurements within the mixing layer (except in the second profile for heights between ~500 to ~1000 m). During the first profile the mixing layer height was very low (250 m) and therefore only a small part of the profile was situated within the mixing layer. However, extrapolating the measured values of $N_{CPC}$ and $N_{CCN,mCCNc}$ in the lowest available altitude to the ground leads to a good agreement with the respective ground-based measurements at Melpitz at 09.50 and 10.30 UTC for the PNC and 10.20 UTC for CCN-NC

respectively.

In the second profile of $N_{CPC}$ and $N_{CCN,mCCNc}$ (each blue) two distinct layers in a height of around 600 m and between 800 and 1100 m altitude were observed. These layers are characterized by an up to six times higher PNC and up to two times higher CCN-NC than below. The lower layer is located within the PBL at its top, whereas the upper layer is located within the residual layer above the PBL. The higher PNC was caused by a new particle formation event within the residual layer, which also was observed by

Wehner et al. (2010). These new particle formation events can also lead to higher PNC via mixing and entrainment processes at the top of the PBL, which was present at around 600 m altitude (cf. Figure 10b left panel sharp decrease in $N_{CCN,mCCNc}$ derived with the lidar). Below, within the well mixed PBL, the in situ airborne measurements show stable values of the PNC ($N_{CPC}$ of around 1800



cm$^{-3}$) and the CCN-NC ($N_{CCN,mCCNc}$ of around 700 cm$^{-3}$). We furthermore assume that the larger CCN-NC were caused by mixing processes with the residual layer at the top of the PBL. An increase in the ground-based CCN-NC was not observed.

We conclude that ground-based measurements can be representative for the PBL, especially in its well-mixed state. However, local events, like new particle formation events in the residual layer or at the top of the PBL as well as pollution plumes near the ground

have to be considered. Note that ground-based measurements can represent the PBL in vertical column above only, because spatial variability was observed for the here presented parameters and therefore collocation also has to be considered.

### 4.2. Intercomparison of in situ and lidar-based CCN-NC

Figure 10 shows in the left parts of both panels the CCN-NC derived with the mCCNc on ACTOS (blue and black dots) and derived with the approach of Mamouri and Ansmann (2016) (black and blue solid lines with shaded area). Within the given uncertainties of

the lidar-based approach (factor two, shaded area), the in situ measurements agree with the lidar-based approach, especially within the planetary boundary layer, since the uncertainty range (shaded area) covers almost all mCCNc data points. Above the PBL the agreement is less distinct, especially in the case of flight 27a for both profiles. In this case we assume that a different aerosol type is prevalent so that the approach of Mamouri and Ansmann (2016) for continental aerosol is not entirely applicable for the investigated altitudes.

Figure 12 shows the correlation of the airborne-based and lidar-based CCN-NC. All data points were derived for five profiles conducted during the three flights (14a (1 profile), 14b (2 profiles) and 27a (2 profiles)) and were logarithmized to prevent an overrepresentation of data clusters. The data points were correlated for altitudes above 350 m. For each measurement of the mCCNC, the respective CCN-NC (same altitude as ACTOS at this moment) was taken from the respective smoothed lidar profile (see Figure 10 and Table 4) from its respective altitude. Table 4 shows the start and end time of the airborne profiles as well as the averaging

period of the respective lidar profiles. The given error-bars assign the given uncertainty of the lidar approach of factor 2 and the assumed uncertainty of 10 % for the mCCNC measurements.

On average the CCN-NC derived from the lidar fit to the airborne CCN-NC measurements (fit with slope 0.994) with a high correlation coefficient of 0.977. This shows on the first glance, that this approach is a feasible instrument to evaluate CCN-NC profiles with remote sensing. On the second glance, in Figure 12 its clearly visible, that the lidar approach overestimates the airborne

CCN-NC measurements for values of $\log_{10}(N_{CCN,mCCNc})$ from 2.7 cm$^{-3}$ to 3.4 cm$^{-3}$ (500 to 2500 cm$^{-3}$ in real conditions) by a factor of two, whereas in the range from $\log_{10}(N_{CCN,mCCNc})$ = 1.8 to 2.5 cm$^{-3}$ (60 to ~320 cm$^{-3}$ real concentrations) the lidar approach underestimates. This indicates different aerosol types and explains the low correlation. Note, in the regime up to $\log_{10}(N_{CCN,mCCNc})$ = 1.8 cm$^{-3}$ the lidar approach overestimates the mCCNc measurements up to a factor of 5. In this cases we assume that the aerosol loading is too low for a reliable retrieval of CCN-NC.

We used Eq. (7) and (8) to derive the CCN-NC from the lidar measurements. This equations were derived for continental aerosol. However, the characterization of the aerosol is important since an aerosol layer above the PBL might have different microphysical and chemical properties. Furthermore, the horizontal inhomogeneities are not entirely captured by the lidar but can be resolved by ACTOS. The shortest duration of one ACTOS profile was 6 minutes (flight 27a, profile 2; see Table 4). With its true air speed of 20 m s$^{-1}$ ACTOS passes a horizontal distance of about 7.2 km within this period. Therefore, parts of the profile were not flown

within the field of view of the lidar and therefore the lidar might not capture aerosol features observed by ACTOS.



### 4.3 Intercomparison of optical parameters

Exemplarily, a comparison of in situ based calculated and lidar-observed profiles of aerosol optical properties for three legs of flight 14b will be presented in this section. Afterwards, a summary of all investigated horizontal legs is given.

### 4.3.1 Case study of flight 14b – Legs D, E and F

The comparison of the profiles of the lidar-based ($\sigma_{\text{bsc,lid}}$) and in situ airborne-based calculated particle backscatter coefficient ($\sigma_{\text{bsc,mie}}$) for the horizontal legs D to F of flight 14b is given in Figure 13. The figure shows a composite of the vertical profile of $RH$, $T$, (panel a)) $N_{\text{CPC}}$ and $N_{\text{OPSS}}$ (panel b)) composed from the vertical, ascending parts before leg A and between leg A and B. The profiles of $\sigma_{\text{bsc,lid}}$ and the mean and three times standard deviation (dots with error bars) of $\sigma_{\text{bsc,mie}}$ are shown in panel c). The solid colored lines represent the measurement at 355, 532 and 1064 nm of the lidar. The transparent filled areas around the solid lines with the respective

color represent the 15% uncertainty for the measurements of the lidar. The measured profile of $\sigma_{\text{ext,lid}}$ and $\sigma_{\text{ext,mie}}$ for the three lidar wavelengths is shown in the panel d) (same color setup as in panel three). The dashed colored lines around the respective $\sigma_{\text{ext,lid}}$ profiles represent the profiles of $\sigma_{\text{ext,lid}}$ derived with the backscatter-to-extinction conversion ($LR$ of 55 sr for 355 and 532 nm and $LR$ of 30 sr for 1064 nm; see Sec. 2.3) with a $LR \pm 15$ sr larger and smaller, respectively. The colors correspond with the colors of the lidar wavelengths.

The profiles of $T$ and $RH$ show an inversion at approximately 1200 m. This inversion is also characterized by a sharp decrease in both $N_{\text{CPC}}$ and $N_{\text{OPSS}}$ by a factor of 12 and 8, respectively. The $RH$ drops from around 85 % to 50 %. Up to this height the layer is characterized by a steady increase of the $RH$ from 45 % to 85 %. Below the inversion, up to a height of around 330 m $N_{\text{OPSS}}$ is increasing from 18 to 36 cm$^{-3}$. $N_{\text{CPC}}$ shows a high variability in this first part of the profile, maybe due to helicopter exhausts released during the ascent of the helicopter. Above the height of 330 m $N_{\text{CPC}}$ and $N_{\text{OPSS}}$ show a value of around 10000 to 13000 cm$^{-3}$ and 40

to 45 cm$^{-3}$, respectively, whereby $N_{\text{CPC}}$ is 30 % larger at the top of the mixing layer then in 330 m. The lidar profiles of $\sigma_{\text{bsc}}$ and $\sigma_{\text{ext}}$ show the same behavior. For each of the three investigated wavelengths $\sigma_{\text{bsc,lid}}$ and $\sigma_{\text{ext,lid}}$ increase up to a height of 1100 m followed by a decrease up to a height of 1500 m. In contrast to the sharp decrease in the first both panels, presenting a "snapshot" of the atmosphere, the slight, smooth decrease of the measured optical coefficients at the top of the mixing layer results from the averaging of the lidar measurements between 13.15 and 13.30 UTC. In this period the mixing layer is still developing (see Figure 3) and thus

the layer with an increased PNC is still growing.

Figure 13 shows a clear correlation of $RH$ and the particle light backscatter and extinction coefficient. While $N_{\text{OPSS}}$ is almost constant between 330 and 1100 m altitude, $RH$ increases with height and due to the hygroscopic growth the cross section (more surface of the particles scatters and absorbs more light) of the aerosol particles increases as well. Quantitatively, the Mie-calculation also produces larger particle light backscatter and extinction coefficients (dots with error bars in panel c)) under conditions with an

elevated $RH$. The conditions during leg E (smaller $RH$) led to a smaller $\sigma_{\text{bsc,mie}}$ and $\sigma_{\text{ext,mie}}$ than during leg D and F.

Figure 13 shows that in contrast to the particle light extinction coefficient, $\sigma_{\text{bsc,mie}}$ is smaller than $\sigma_{\text{bsc,lid}}$ for each leg and wavelength. Even within the considered uncertainties measurements and model do not agree with each other. An underestimation of $\sigma_{\text{bsc}}$ by the Mie-calculation may result from the non-detected particle size range between 226 and 356 nm, so that the integration method (trapezoidal) may underestimates the particle light backscatter coefficient in this size-range. More importantly though, we expect

that the setup used on ACTOS does not detect particles with an optical diameter larger than 2.8 µm and thus those particles are not considered in the Mie-calculations as well. Particles with a diameter about six times larger than the incoming wavelength are most





effective in backscattering (Figure A1, largest backscatter efficiency at a size parameter of 19). For the here investigated wavelengths particles with a diameter of ~2 µm (355 nm), ~3.2 µm (532 nm) and ~6 µm would be most effective. Especially, for 532 and 1064 nm these particles were not detected with the used setup. In contrast, the lidar detects all aerosol particles. Additionally, the horizontal distance between the lidar and ACTOS could be a reason, because the columnar measurements of the lidar just partly match with the flight-pattern of ACTOS. Finally, the lidar resolves the horizontal inhomogeneity in the atmosphere (see Figure 9) columnar. Therefore, features of these horizontal inhomogeneities can be detected by ACTOS and the lidar at different times. This might explain disagreements of the Mie-calculations and the lidar measurements.

Because of the constant multiplication of $\sigma_{bsc,lid}$ with the $LR$, the general behavior of the particle light extinction measured by the lidar does not differ from the backscatter measurements. The relative difference between the extinction at 1064 nm to 532 and 355 nm is different due to the smaller $LR$. For leg D and leg E, both located roughly 1000 m above ground, $\sigma_{ext,mie}$ coincides with $\sigma_{ext,lid}$ for $\lambda = 355$ and 532 nm. The $\sigma_{ext,mie}$ of leg E is for all wavelengths smaller than the lidar-based $\sigma_{ext}$. For 355 nm $\sigma_{ext,mie}$ is 44 %, for 532 nm 38 % and for 1064 nm 53% smaller than $\sigma_{ext,lid}$. A smaller $LR$ could explain this discrepancy, but $LR_{mie}$ for 355 and 532 nm is larger than the here used 55 sr, which is explained by the underestimation of $\sigma_{bsc}$ by the Mie-calculations. Clean marine aerosol as stated in Bréon (2013) provides a $LR$ of around 25 sr for 670 nm, which is slightly larger than in the study of Groß et al. (2011), who found that a transported clean marine aerosol (measured at Praia, Cape Verde Islands) causes slight wavelength depending $LR's$ of 14 to 24 sr at 355 nm and 17 to 19 sr at 532 nm wavelength. Also, Groß et al. (2011) showed that a mixture of biomass-burning aerosol and dust is characterized by a wavelength independent $LR$ of 57 to 98 sr for 532 and 355 nm. Based on 10 years of Raman lidar observations in Europe, Asia and Africa Müller et al. (2007) characterized the $LR$ for several aerosol types within the PBL or in the free troposphere (FT). For 532 nm lidar systems within the PBL lidar ratios were found between 23 ± 3 for a marine aerosol and 55 ± 5 sr for mineral dust of the Sahara. For 355 nm they found lidar ratios between 55 ± 6 sr for mineral dust of the Sahara and 58 ± 12 sr for urban or anthropogenic haze aerosol in Central Europe. The investigations for central Europe are of special interest, because they are representative for the Raman lidar data set used here. In this case they found a $LR$ of 53 ± 11 sr for 532 nm and 58 ± 12 sr for 355 nm. Omar et al. (2009) present a satellite-based study, which provides model-based lidar ratios for different aerosol-types for 532 and 1064 nm. For the cases of clean continental, polluted continental and polluted dust the lidar ratios for 1064 nm were 30 sr.

### 4.3.2 Discussion of backscatter coefficient closure

Table B1 lists the values of each $\sigma_{bsc}$ data point that we investigated in this study and which are shown in Figure 14. For the lidar measurements the 15% error and for the calculated values the three times standard is shown. For fields marked with "-" we got negative values of $\sigma_{bsc,lid}$ because of the low aerosol loading in combination with the measurement uncertainties. We neglected these negative values because they are physically not reasonable.

For each flight, $\sigma_{bsc}$ is larger within the mixing layer than above. For flight 14b and 27a at least two legs were located within the mixing layer. During flight 14b $\sigma_{bsc}$ shows a low variation within the mixing layer. The $\sigma_{bsc}$, derived from lidar measurements during leg D and F, performed at the same heights (999 ± 16 and 1006 ± 17 m), varying within 5 % (see Tab. B1). $\sigma_{bsc,lid,355}$ of leg E, also measured within the mixing layer ($\overline{h_{leg}} = 382$ m), is around 20 % smaller than in leg D and F. This can be explained by an enhanced hygroscopic growth due to the higher $RH$ at around 1000 m than at 380 m (see Figure 13). In contrast, the $\sigma_{bsc,lid}$ of leg B is around five times higher than that derived for leg D of flight 27a, although measured in the same height. This can be explained by the





difference in the time of measurement of around 65 min. While in leg B most of the aerosol mass is trapped within the mixing layer in the lowest 300 m, the thickness of the ML reached approximately 750 m height at the measurement time of leg D. Due to turbulent mixing of cleaner air from above the ML trapped aerosol gets diluted an therefore the PNC decreased. At 532 nm wavelength the leg B of flight 14a shows larger values of $\sigma_{bsc,lid}$ than at 355 nm. Due to the low value of the measured $\sigma_{bsc}$ this could be explained by the measurement uncertainty.

Figure 14 shows the correlation between the calculated and the measured $\sigma_{bsc}$ of all investigated legs. The error bars represent the considered uncertainties of the lidar and the three times standard deviation of mean of the Mie-algorithm. According to flight-time and mean flight-height of the horizontal legs, we choose the respective lidar-profile to compare the Mie-based values with the lidar-profiles at the respective height (see Figure 13). The lidar value at the respective height was derived by linear interpolation between to height-steps of the lidar profiles. For all wavelengths, $\sigma_{bsc,mie}$ shows on average smaller values than $\sigma_{bsc,lid}$. For measurements at a wavelength of 355 and 532 nm, values of $\sigma_{bsc,mie}$ are about 30% smaller (355 nm 29.9 % and 532 nm 27.6 %) and for $\lambda = 1064$ nm 50 % smaller. This results might be mainly due to the fact that particles most efficient in backscattering were not observed with the airborne setup.

In addition, it is clearly visible that on each flight the backscatter coefficients are smaller above the PBL (empty symbols) compared to those within the PBL (filled symbols). This is caused by the lower aerosol concentration above the mixing layer.

Summarizing, 4 out of the 13 (30.8 %) considered cases agree (cf. Table B1) for 355 nm, 4 out of 11 for 532 nm (36.4 %), and 5 out of 13 for 1064 nm (38.5 %) lidar measurements match with the particle light backscatter coefficients based on the airborne in-situ measurements. Still, the conversion from in-situ measurements to $\sigma_{bsc}$ is possible with the underlying assumptions and partly agrees with direct measurements of the here used lidar system. Nevertheless, an improved measurement set-up is certainly needed. Especially within the PBL, the determination of the PNSD is important as stated by Kent et al. (1983), especially for particles larger than 1 µm, although they considered $\sigma_{bsc}$ for light of a wavelength of 10.6 µm. With our set-up, we cover particles up to a size of 2.8 µm in optical diameter. The upper cut-off of the inlet-system was unfortunately at about 2 µm. In contrast, the lidar system detects all particle sizes. Therefore, prevalent particles with a diameter larger than the upper detection limit of the airborne in situ instrumentation are not considered in the optical calculation and so the backscattering is underestimated by the Mie-algorithm. An OPSS with a larger detection range as well as larger upper sampling cut-off of the inlet could overcome this problem.

### 4.3.3 Discussion of the extinction coefficient closure

**355 and 532 nm wavelength**

Considering all measurement points of this study, the particle light extinction coefficient shows a different behavior than the particle light backscatter coefficient converted from the aerosol in situ measurements, which is significant smaller than the lidar derived particle light backscatter coefficient. In Figure 15, the correlation of $\sigma_{ext,lid}$ and $\sigma_{ext,mie}$ is shown (error bars are the same as in Figure 14). $\sigma_{ext,lid}$ and $\sigma_{ext,mie}$ agree within 8% with each other with a high correlation coefficient $R^2$ of 0.944 for 355 and 0.947 for 532 nm respectively. For $\lambda = 355$ nm the Mie-algorithm calculates on average 3.5 % smaller values than the lidar. This implies, that the here used $LR$ for 355 and 532 nm are valid. In contrast, the calculated particle light extinction coefficient is overestimated compared to the lidar-based particle extinction on average by 7.9 % for 532 nm. According to the values in Tab. B2 7 out of 13 (54%) $\sigma_{ext,mie}$ values agree with the measured $\sigma_{ext}$ for 355 nm and for 532 nm 6 out of 11 (55%) $\sigma_{ext,mie}$ agree with the measured lidar-based ones. While Groß et al. (2011) found wavelength independent $LR$ for 355 and 532 nm, the here used algorithm produces different $LR$ for





the different wavelengths and horizontal legs, especially for 1064 and 355/532 nm. On average, the $LR$ at 355 and 532 nm is 69.9 $\pm$ 13.3 and 70.9 $\pm$ 21.2, respectively. An underestimation of $\sigma_{bsc}$ due to the in situ sampling setup has to be considered and so these $LR$ might be too high. Nevertheless, these $LR$ agree with Groß et al. (2011), and furthermore, the $LR_{mie}$ of around 70 sr agrees with a $LR$ of 58 $\pm$ 12 for 355 nm as given in Müller et al. (2007).

**1064 nm wavelength**

The scatter plot of $\sigma_{ext,lid}$ and $\sigma_{ext,mie}$ for 1064 nm is given in Figure 16. On average, the algorithm calculates 5.2 % smaller values than derived by the lidar, but compared to 355 and 532 nm $R^2$ is significant smaller (0.77). In the range of 0 to 20 Mm$^{-1}$ in $\sigma_{ext}$ for 1064 nm, the correlation is close to the 1:1 line (black solid line). Above this range, the correlation is less significant. The measurements and the Mie-calculations have a significant difference for leg D of flight 14b (see Figure 16, $\sigma_{ext,mie}$ = 61.00 $\pm$ 16.22

Mm$^{-1}$, $\sigma_{ext,lid}$ = 23.73 $\pm$ 3.56 Mm$^{-1}$). This might be caused by an overestimation of the PNSD, especially for the first two channels of the OPSS, which causes an overestimation of the total particle cross section in this size range. The overestimation of $\sigma_{ext}$ is also clearly visible for the wavelengths 355 and 532 nm. For leg D and F of flight 14b the Mie-based values are 35% (leg D) and 42% (leg F) smaller than derived by the lidar. Within the range of the $LR$, the in situ and lidar-based particle light extinction coefficients coincide. The Mie-based $LR_{mie}$ is 19.6 and 20.3 sr for leg D and leg F, respectively. By using these values for calculation $\sigma_{ext,lid}$ from

$\sigma_{bsc,lid}$ the $\sigma_{ext,lid}$ becomes 28.42 (leg D) and 29.43 Mm$^{-1}$ (leg F). This agrees with a $\sigma_{ext,mie}$ of 28.31 $\pm$ 4.87 Mm$^{-1}$ derived during leg D and 25.75 $\pm$ 8.03 Mm$^{-1}$ for leg F. For 1064 nm and leg E of flight 14b the $LR_{mie}$ is 17.1 sr. Using this $LR$ $\sigma_{ext,mie}$ and $\sigma_{ext,lid}$ agree with each other within the uncertainties. According to the values in Tab. B2 and for 1064 nm 7 out of 13 (54%) $\sigma_{ext,mie}$ agree with $\sigma_{ext,lid}$ using a $LR$ of 30 sr. A summary of all investigated data points of $\sigma_{ext}$ for all three investigated wavelengths is given in Tab B2.

**4.3.4 Influence of a different $\kappa$ measurement**

Kristensen et al. (2016) described a method to derive the hygroscopicity based on PNSD and total CCN-NC measurements (here with the mCCNc on ACTOS) at a certain supersaturation. Applied to the airborne data-set used here, non-reliable values of the particle hygroscopicity with a high standard-deviation were ascertained. Although the CCN-NC seems to be very stable with height and time during the day (see Figure 10), the method of Kristensen et al. (2016), which is based on the evaluation of the critical diameter is very sensitive to the PNSD. The size-resolution, the low counting statistic, but mainly the non-observed size-range in

the PNSD (between 226 and 356 nm), derived with the MPSS and OPSS on ACTOS leads to high variations in the calculated critical diameter and thus a variation in the particle hygroscopicity resulting in unreasonable high or low hygroscopicity parameters. Measurements of CCN-NC are available at ground and overall they are representative for higher altitudes (see Figure 10), but their temporal resolution is lower than that of the ground-based chemical measurement. Hence, we decided to use the volume-weighted mixing approach based on chemical measurements on ground to calculate the hygroscopic growth of the aerosol particles.

Anyhow, based on the ground-based CCN-NC measurements, the hygroscopicity of the aerosol particles can be also derived. The derived $\kappa$ is evaluated by using the method described in Sect. 2.1.3 and the results of the influence of the hygroscopicity Mie-based airborne aerosol optical properties is given in the following. A comparison of the $\kappa$ derived with the Q-ACSM and CCNc measurements on ground are given in Table 5. For both days of interest, the CCNc on ground led to smaller values than that derived by the Q-ACSM. This might be due to the rough assumption of a constant particle composition. Furthermore, the compounds of the



aerosol particles where measured for $PM_1$, whereby the CCNc measured $\kappa$ for $PM_{10}$. Aerosol particles with a smaller hygroscopicity and a diameter larger than 1 µm will not be detected by the Q-ACSM and so the Q-ACSM measurements might lead to an overestimation of $\kappa$. In addition, water-insoluble aerosol species are not detected by the Q-ACSM. This water-insoluble aerosol compounds have a smaller hygroscopicity than the soluble compounds. Therefore, the chemical composition approach might

overestimate $\kappa$. On the other hand, the CCNc measurements might over- or underestimate $\kappa$ which is only valid in the size-range of the derived critical diameter.

A comparison of the correlation of the Mie-based aerosol optical properties derived with both approaches, the chemical composition and the CCNc-based, and the lidar-based aerosol optical properties is given in Table 6. Table 6 provides parameters describing the correlation function $\sigma_{mie} = a\ \sigma_{lid}$ with its respective correlation coefficient $R^2$ for the Mie-calculations using the $\kappa$ based on both

approaches.

Compared with the chemical composition approach, the hygroscopicity taken from the ground-based CCN-NC measurements (see Table 5) leads to smaller ambient state optical properties (see Table 6). This is caused by a lower simulated growth of the aerosol particles due to the smaller hygroscopicity and therefore a lower cross-section of the grown aerosol particles.

The general assumption of a constant $\kappa$ over all sizes in both approaches may not be justified. Size-resolved $\kappa$ might reduce the

errors in the simulation of the hygroscopic growth and so reduces the uncertainties in the aerosol optical properties. A more satisfying approach would be to apply size-resolved measurements of the aerosol particle growth factor (GF) or hygroscopicity on the derived airborne PNSD since the chemical composition of the aerosol particle varies with their size depending on the origin of the aerosol particles.

### 5 Summary and Conclusions

To investigate optical properties of aerosol particles in ambient state, an intensive field study was conducted as part of the HD(CP)² Observational Prototype Experiment HOPE at the Central European research observatory Melpitz, Germany. Aerosol particle light backscatter and extinction coefficients, based on vertical and horizontal, highly spatiotemporally resolved aerosol measurements, have been compared to profiles of such aerosol optical properties at three wavelengths derived with remote-sensing instruments. To be able to do this, the hygroscopic growth of aerosol particles was simulated using the hygroscopicity parameter $\kappa$ derived from

ground-based chemical composition and CCN-NC measurements.

In this study, ground-based measurements of the PNC were found to be not always representative for higher atmospheric layers within the planetary boundary layer. In particular, new particle formation events in the residual layer (Wehner et al., 2010) can lead to a higher PNC and vertical variation inside the PBL. These elevated aerosol PNCs are not connected with ground-based measurements. On the other hand, transported air masses on ground with a higher PNC can be decoupled from higher atmospheric

layers and so the ground-based measurements also do not entirely represent elevated atmospheric layers - at least in the here presented cases. Nevertheless, in a well-mixed PBL, ground-based measurements provide a good estimate of the aerosol particle properties within the PBL.

The CCN-NC was also variable within the developing planetary boundary layer since entrainment processes at the top of the planetary boundary layer can led to an increased CCN-NC, especially close to the top of the PBL, which was not captured by ground-

based measurements.



For three investigated flights, profiles of logarithmized (base 10) CCN-NC derived with the approach of Mamouri and Ansmann (2016) were compared with airborne in-situ measurements of CCN-NC (logirthmized with base 10) and showed a surprisingly good agreement within 1 % (lidar approach is lower) with a correlation coefficient of 0.977. Although different supersaturations have been considered (0.2 % in-situ and 0.15 % lidar-approach) and the lidar-based approach of Mamouri and Ansmann (2016) underlies

uncertainties of a factor of about two, the approach is a helpful tool to evaluate CCN-NC with the lidar.

Furthermore, comparisons of Mie-theory-based and lidar-based particle light backscatter coefficients implies that the here used setup cannot provide a complete data-base to reproduce the "real" particle light backscatter coefficient since the investigated size-range seems to be too small. This can be mainly explained by the complex behavior of the backscatter efficiency of aerosol particles in the narrow scattering angle window in 180° direction (cf. Figure A1; high backscatter efficiency of particles ~6 times larger in

diameter than the incoming radiation).

Within the uncertainty ranges, the particle light backscatter coefficients on the basis of the airborne in-situ measurements agree with the measured $\sigma_{bsc}$ in up to 38.5 per cent of the cases. On average, the algorithm used here retrieves 29.9, 27.6 and 50 % smaller $\sigma_{bsc}$ compared to the measured ones at 355, 532 and 1064 nm. In contrast, the conversion from airborne in situ aerosol measurements to $\sigma_{ext}$ yields promising results. For 355 and 532 nm, the Mie-based $\sigma_{ext}$ reproduces the measured $\sigma_{ext}$ within 8 % deviation and with a

high correlation coefficient ($R^2 > 0.94$). Nevertheless, this study shows, that the aerosol-type dependent intensive property of the $LR$, which is assumed as height independent during day times in many publications, leads to uncertainties in particle light extinction profiles derived from backscatter profiles, since the here used algorithm provided different $LR$ in different heights. However, on average a $LR$ of 55 sr for 355 and 532 nm is applicable for the here-investigated aerosol-type.

In contrast, the knowledge of $LR$ at 1064 nm is rare from direct active lidar measurements. First measurements to evaluate the $LR$ at

1064 nm have been done by Haarig et al. (2016) with a rotational Raman-lidar for a cirrus cloud case. In this cirrus case, they derived a $LR$ for 1064 nm of 38 ± 5 sr. The here presented study shows that a $LR$ of 30 sr provides on average a good agreement between Mie-based and lidar-based $\sigma_{ext}$ for the presented cases. This is also shown in the model-based study of Omar et al. (2009) for clean and polluted continental and polluted dust aerosol ($LR$ of 30 sr). However, the algorithm used here provided an average $LR$ for 1064 nm of 15.8 ± 6.7 sr (3.8 and 28.1 minimum and maximum).

An investigation of the representativeness and the optical closure study would be interesting for more polluted and stable cases, since the lidar-measurements with coupled Sun-photometer measurements provide more robust results under these conditions (Dubovik et al., 2002, Bond et al., 2013).

### Appendix A: Effectiveness of Mie-Scattering

Mie-scattering is most effective for particles in the size range of the wavelength of the incoming radiation. The ratio of particle size

($D_p$) and the wavelength of the incoming electromagnetic radiation ($\lambda$) multiplied with $\pi$ is described as size parameter $x$. This parameter is defined as:

$$x = \pi \frac{D_{\mathrm{p}}}{\lambda}. \tag{A1}$$

Figure A1 shows the extinction, scattering, absorption and backscatter efficiency $Q_{ext,sca,abs,bsc}$ depending on the size parameter $x$ for spherical layered particles. They consist of a core of eBC (volume fraction of 0.05) and a shell of less-absorbing water-soluble





material. The refractive index of eBC and the less absorbing material were taken from Table 3. For scattering, extinction and absorption the maximum in the efficiency is reached for an $x$ of around 3. According Eq. (A1) this means the ratio of $D_p$ and $\lambda$ is unity. The scattering efficiency narrows unity with an increase of $x$. In contrast, the backscatter efficiency is maximal for an $x$ of 19. In contrast to the other optical efficiencies $Q_{bsc}$ shows a higher variability. Therefore a precise calculation of $Q_{bsc}$ needs a very precise

5    determination of the PNSD. Compared to the extinction efficiency the ripple structure in the backscatter efficiency is more distinct. Therefore $\sigma_{ext}$ is less sensitive to the particle counting and sizing. Especially for aerosol particle in the size range of the OPSS detection limit are these ripples are of major interest.

**Appendix B – Tables of derived and measured optical coefficients**



**Acknowledgements**

Grateful we are for the competent help of the technicians Thomas Conrad and Astrid Hofmann and we thank all the other employees of the TROPOS, who have supported us so energetically and courageously before, during, and after the campaign, and have made the campaign a full success. Furthermore, we are very thankful to the helicopter pilots Alwin Völlmer and Jürgen Schütz of the
5      Rotorflug airservices GmbH & Co. KGaA for the secure flights. The authors furthermore thank Dieter Schell of enviscope GmbH for his expertise. Furthermore, we are thankful to Gregory C. Roberts of Scripps Institution of Oceanography (Center for Atmospheric Sciences, La Jolla, USA) for providing the custom-built mini cloud condensation nuclei counter. This study was mainly carried out in the project HD(CP)$^2$ funded by the German Ministry for Education and Research. We specifically acknowledge the HD(CP)$^2$ project 01LK1212C (TROPOS). The authors gratefully acknowledge the NOAA Air Resources Laboratory (ARL) for the
10    provision of the HYSPLIT transport and dispersion model and/or READY website (http://www.ready.noaa.gov) used in this publication.



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





**Tables and figures**

**Table 1: Hygroscopicity $\kappa$ and density $\rho$ of each considered aerosol particle compound**

| Component | $\kappa_i$ | $\rho_i$ [g cm$^{-3}$] |
|---|---|---|
| eBC | 0[1] | 1.5[4] |
| Organics | 0.1[2] | 1.27[3] |
| NH$_4$NO$_3$ | 0.67[2] | 1.735[5] |
| H$_2$SO$_4$ | 0.9[2] | 1.84[5] |
| NH$_4$HSO$_4$ | 0.61[2] | 1.78[5] |
| (NH$_4$)$_2$SO$_4$ | 0.61[2] | 1.76[5] |

[1]Wu et al., (2013); assumed to be 0
[2]Zaveri et al., (2010)
[3]Ma et al., (2014)
[4]Cross et al., (2007)
[5]Lin et al., (2014)

**Table 2: Summary of take-off and landing times of the respective flights of HOPE.**

| flight | take-off | landing |
|---|---|---|
| [yyyymmdd a/b] | [UTC] | [UTC] |
| 20130912a | 13.02 | 13.41 |
| 20130913a | 08.51 | 10.36 |
| 20130914a | 08.19 | 10.16 |
| 20130914b | 12.05 | 13.54 |
| 20130917a | 08.36 | 10.31 |
| 20130921a | 11.15 | 13.07 |
| 20130922a | 08.56 | 10.48 |
| 20130927a | 08.08 | 10.10 |

**Table 3: Real ($n_{re}$) and imaginary part ($n_{im}$) of the complex refractive index ($n$) of the aerosol components used for the volume weighted mixing in the algorithm to derive $n$ of the core and the shell of the aerosol particles. Also the standard deviation ($\sigma$) of each part of $n$ is given in which the algorithm varies the $n$ of each compound to provide a possible range of values for $\sigma_{ext}$ and $\sigma_{bsc}$. The values in this table where taken out of Ma et al. (2014).**

| component | $n_{re}$ | $\sigma(n_{re})$ | $n_{im}$ | $\sigma(n_{im})$ |
|---|---|---|---|---|
| soluble | 1.53 | 0.5% | 1e-06 | - |
| eBC | 1.75 | 4% | 0.55 | 6.6% |
| water | 1.33 | 0.5% | 0 | - |



**Table 4: Overview of start and end time of the profiles conducted during the flights 14a, 14b, and 27a. Also the averaging period for the respective lidar profiles is given.**

|  |  | ACTOS |  | Lidar |  |
| --- | --- | --- | --- | --- | --- |
| flight | profile | Start [UTC] | End [UTC] | Start [UTC] | End [UTC] |
| 14a | 1 | 08.50 | 09.26 | 08.50 | 09.24 |
| 14b | 1 | 12.05 | 12.27 | 12.20 | 12.39 |
| 14b | 2 | 13.47 | 13.54 | 13.15 | 13.29 |
| 27a | 1 | 10.08 | 10.34 | 10.05 | 10.27 |
| 27a | 2 | 11.29 | 11.35 | 11:25 | 11.57 |

5  **Table 5: Mean $\kappa$ for the ground-based measurements of the CCNc and Q-ACSM recorded between 08.00 and 15.00 UTC for the here investigated days.**

| date | $\kappa_{CCNc}$ | $\kappa_{Q\text{-}ACSM}$ |
| --- | --- | --- |
| 2013-09-14 | $0.33 \pm 0.05$ | $0.43 \pm 0.03$ |
| 2013-09-27 | $0.24 \pm 0.06$ | $0.41 \pm 0.03$ |

**Table 6: Parameters of the correlation of the Mie-based and the lidar-based optical properties. Hygroscopicity derived on the basis of the chemical composition and CCNc measurements on ground. $a$ is the slope of the linear fit and $R^2$ is the correlation coefficient.**

|  | $\lambda$ [nm] | extinction | | backscattering | |
| --- | --- | --- | --- | --- | --- |
|  |  | $a$ | $R^2$ | $a$ | $R^2$ |
| composition based | 355 | 0.965 | 0.944 | 0.701 | 0.925 |
|  | 532 | 1.079 | 0.947 | 0.724 | 0.953 |
|  | 1064 | 0.948 | 0.796 | 0.5 | 0.819 |
| CCN based | 355 | 0.829 | 0.952 | 0.583 | 0.933 |
|  | 532 | 0.908 | 0.958 | 0.589 | 0.975 |
|  | 1064 | 0.757 | 0.777 | 0.451 | 0.782 |





**Table B1:** Table shows the aerosol particle light backscatter coefficient ($\sigma_{bsc}$) derived with the lidar for the wavelengths 355, 532, 1064 nm for the mean height of each investigated leg with the 15 % error. Also, the aerosol particle light backscatter coefficient from the airborne in situ aerosol measurements is printed for the respective lidar wavelength and horizontal leg with the three times standard deviation of mean of the 50 runs of the Monte-Carlo simulation. Additionally, it is shown if the horizontal flight leg was conducted within or above the PBL. Bold written values representing a disagreement between lidar and model, normal an agreement. Brackets around the values signing no lidar measurements available.

| | | | | lidar $\sigma_{bsc}(\lambda) \pm 15\%$ [Mm$^{-1}$sr$^{-1}$] $\lambda$ | | | Mie-based $\sigma_{bsc}(\lambda) \pm 3\sigma_{\sigma_{bsc(\lambda)}}$ [Mm$^{-1}$sr$^{-1}$] $\lambda$ | | |
| flight | leg | PBL | $h_{leg}$ [m] | 355 nm | 532 nm | 1064 nm | 355 nm | 532 nm | 1064 nm |
|---|---|---|---|---|---|---|---|---|---|
| 14a | A | no | 605 | **2.31 ± 0.35** | **1.25 ± 0.19** | 0.30 ± 0.04 | **1.10 ± 0.22** | **0.53 ± 0.11** | 0.32 ± 0.07 |
| 14a | B | no | 1602 | 0.39 ± 0.06 | **0.42 ± 0.06** | **0.11 ± 0.02** | 0.30 ± 0.04 | **0.15 ± 0.02** | **0.08 ± 0.01** |
| 14a | C | no | 994 | **0.78 ± 0.12** | 0.48 ± 0.20 | 0.15 ± 0.02 | **0.54 ± 0.05** | 0.27 ± 0.01 | 0.12 ± 0.01 |
| 14a | D | yes | 378 | 4.52 ± 0.68 | 2.80 ± 0.42 | **0.79 ± 0.12** | 4.71 ± 1.63 | 2.17 ± 0.46 | **1.02 ± 0.09** |
| 14b | A | yes | 366 | **2.55 ± 0.38** | **1.16 ± 0.17** | 0.43 ± 0.06 | **1.38 ± 0.15** | **0.65 ± 0.04** | 0.33 ± 0.03 |
| 14b | B | no | 2244 | - | - | **0.00 ± 0.00** | (0.02 ± 0.00) | (0.01 ± 0.00) | **0.00 ± 0.00** |
| 14b | C | no | 1619 | **0.81 ± 0.12** | - | **0.02 ± 0.00** | **0.24 ± 0.06** | (0.12 ± 0.02) | **0.05 ± 0.01** |
| 14b | D | yes | 999 | **3.73 ± 0.56** | **2.24 ± 0.34** | **1.45 ± 0.22** | **2.09 ± 0.37** | **1.44 ± 0.39** | **0.51 ± 0.07** |
| 14b | E | yes | 382 | **3.05 ± 0.46** | **1.55 ± 0.23** | **0.93 ± 0.14** | **1.48 ± 0.11** | **0.76 ± 0.07** | **0.37 ± 0.02** |
| 14b | F | yes | 1006 | **3.55 ± 0.53** | 2.19 ± 0.33 | **1.45 ± 0.22** | **2.19 ± 0.45** | 1.27 ± 0.67 | **0.62 ± 0.16** |
| 27a | A | no | 372 | **1.06 ± 0.16** | **0.35 ± 0.05** | 0.12 ± 0.02 | **0.34 ± 0.05** | **0.21 ± 0.02** | 0.09 ± 0.04 |
| 27a | B | yes | 195 | 5.43 ± 0.81 | 3.32 ± 0.50 | **1.75 ± 0.26** | 3.95 ± 2.98 | 3.04 ± 0.56 | **0.82 ± 0.26** |
| 27a | C | no | 1559 | **0.41 ± 0.06** | - | - | **0.05 ± 0.02** | (0.03 ± 0.01) | (0.01 ± 0.00) |
| 27a | D | yes | 212 | 1.11 ± 0.17 | **0.77 ± 0.12** | 0.37 ± 0.06 | 1.05 ± 0.32 | **0.49 ± 0.12** | 0.27 ± 0.08 |





**Table B2:** Table shows the aerosol particle light extinction coefficient ($\sigma_{ext}$) derived with the lidar for the wavelengths 355, 532, 1064 nm for the mean height of each investigated leg with the 15 % error. Also, the aerosol particle light extinction coefficient from the airborne in situ aerosol measurements is printed for the respective lidar wavelength and horizontal leg with the three times standard deviation of mean of the 50 runs of the Monte-Carlo simulation. Additionally, it is shown if the horizontal flight leg was conducted within or above the PBL. Bold written values representing a disagreement between lidar and model, normal an agreement. Brackets around the values signing no lidar measurements available.

| | | | | lidar $\sigma_{ext}(\lambda) \pm 15\%$ [$Mm^{-1}sr^{-1}$] | | | Mie-based $\sigma_{ext}(\lambda) \pm 3\sigma_{\sigma_{ext(\lambda)}}$ [$Mm^{-1}sr^{-1}$] | | |
| | | | | $\lambda$ | | | $\lambda$ | | |
| flight | leg | PBL | $h_{leg}$ [m] | 355 nm | 532 nm | 1064 nm | 355 nm | 532 nm | 1064 nm |
|---|---|---|---|---|---|---|---|---|---|
| 14a | A | no | 605 | **126.90 ± 19.04** | **68.84±10.33** | 8.86±1.33 | **80.83±17.99** | **45.20±11.85** | 11.70±3.02 |
| 14a | B | no | 1602 | 21.25 ± 3.19 | **23.21±3.48** | 3.27±0.49 | 21.70±3.697 | **12.14±1.97** | 2.69±0.48 |
| 14a | C | no | 994 | 42.67 ± 6.40 | **26.26±3.94** | 4.57±0.69 | 34.24±2.514 | **17.13±1.15** | 3.74±0.26 |
| 14a | D | yes | 378 | 248.56 ± 37.29 | 153.75±23.06 | **23.73±3.56** | 306.01±64.046 | 216.44±48.09 | **61.00±16.22** |
| 14b | A | yes | 366 | **140.10 ± 21.02** | **63.55±9.53** | 12.88±1.93 | **84.24±8.09** | **47.73±4.33** | 11.79±1.10 |
| 14b | B | no | 2244 | - | - | **0.13±0.04** | (0.94±0.06) | (0.33±0.03) | **0.05±0.01** |
| 14b | C | no | 1619 | **44.58 ± 6.69** | - | **0.72±0.11** | **14.33±3.64** | (6.53±1.24) | **1.22±0.22** |
| 14b | D | yes | 999 | 205.02 ± 30.75 | 123.00±18.45 | 43.50±6.53 | 188.08±27.58 | 113.93±15.62 | 28.31±4.87 |
| 14b | E | yes | 382 | **167.53 ± 25.13** | **85.06 ± 12.76** | **27.96±4.19** | **94.40±6.59** | **52.98±4.18** | **13.07±0.97** |
| 14b | F | yes | 1006 | 195.23 ± 29.29 | 120.64 ± 18.10 | 43.34±6.50 | 206.92±60.67 | 111.66±36.32 | 25.75±8.03 |
| 27a | A | no | 372 | **58.05 ± 8.71** | 18.99 ± 2.85 | 3.49±0.52 | **27.88±10.78** | 13.31±4.20 | 2.61±0.74 |
| 27a | B | yes | 195 | 298.55 ± 44.78 | 182.50 ± 27.38 | 52.41±7.86 | 314.40±64.44 | 218.59±55.49 | 62.15±20.36 |
| 27a | C | no | 1559 | **22.33 ± 3.35** | - | - | **2.46±1.01** | (0.96±0.29) | (0.16±0.06) |
| 27a | D | yes | 212 | 60.99 ± 9.15 | 42.47 ± 6.30 | 11.21±1.68 | 79.97±23.00 | 49.82±16.99 | 12.12±3.90 |





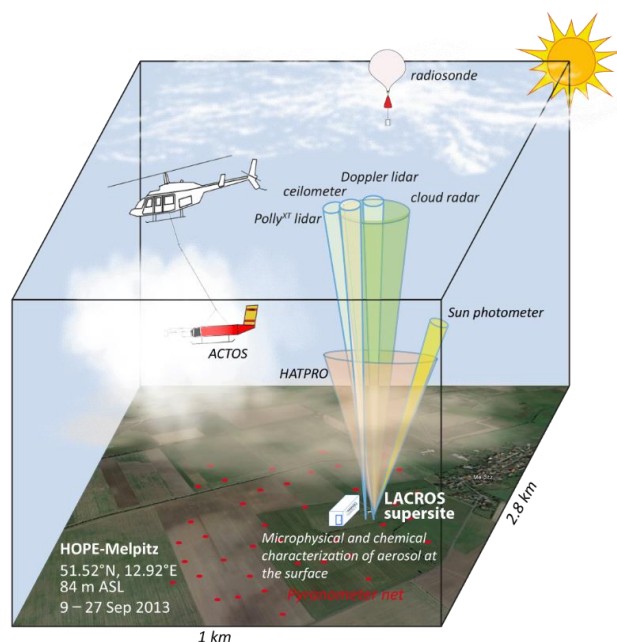

**Figure 1: Scheme of the measurement set-up used during HOPE-Melpitz (from Macke et al. (2017)).**

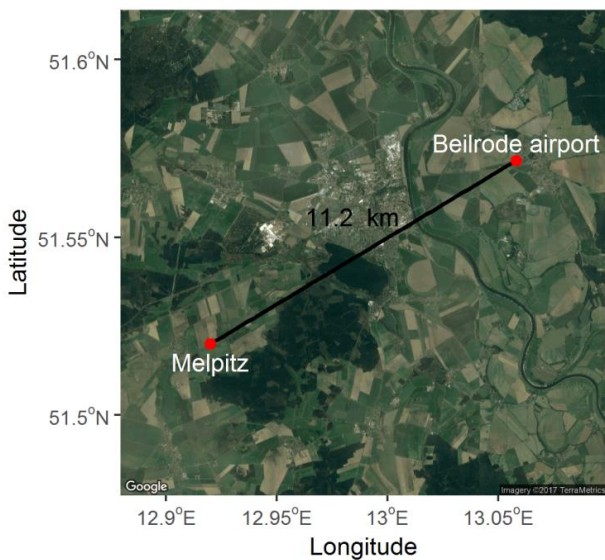

**Figure 2: Location of the measurement sites Melpitz and the airfield in Beilrode. Map from https://www.google.com/maps.**



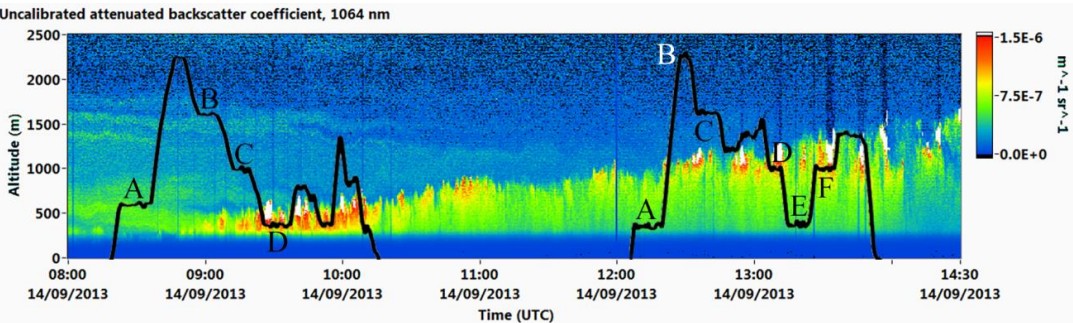

**Figure 3: Range-corrected backscatter signal for 1064 nm derived with Polly$^{XT}$ on September 14. The more reddish, the larger is the backscatter signal. The black lines represent the flight patterns of flight 14a and b. White colors indicate very high backscattering.**

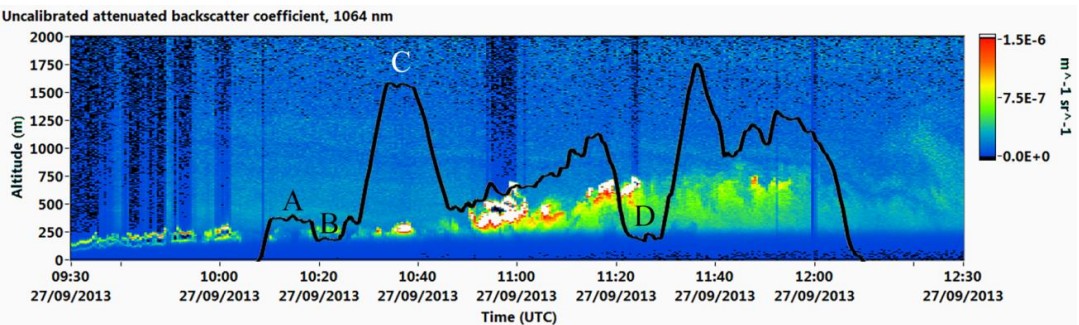

5    **Figure 4: Range-corrected backscatter signal for 1064 nm derived with Polly$^{XT}$ on September 27. The more reddish, the larger is the backscatter signal. The black lines represent the flight patterns of flight 27a. White colors indicate very high backscattering.**





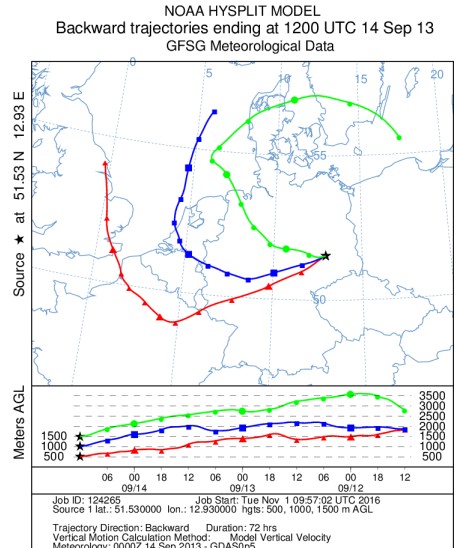

**Figure 5: Three 72h backward trajectories for 500 (red), 1000 (blue), 1500 m (green) above ground for Melpitz ending at Sept. 14, 12 UTC.**

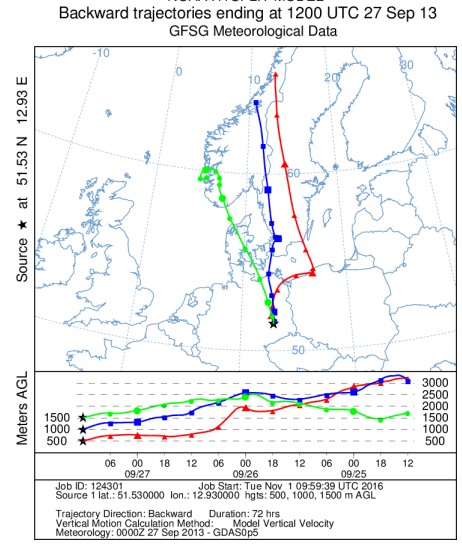

**Figure 6: Three 72h backward trajectories for 500 (red), 1000 (blue), 1500 m (green) above ground for Melpitz ending at Sept. 27, 12 UTC.**





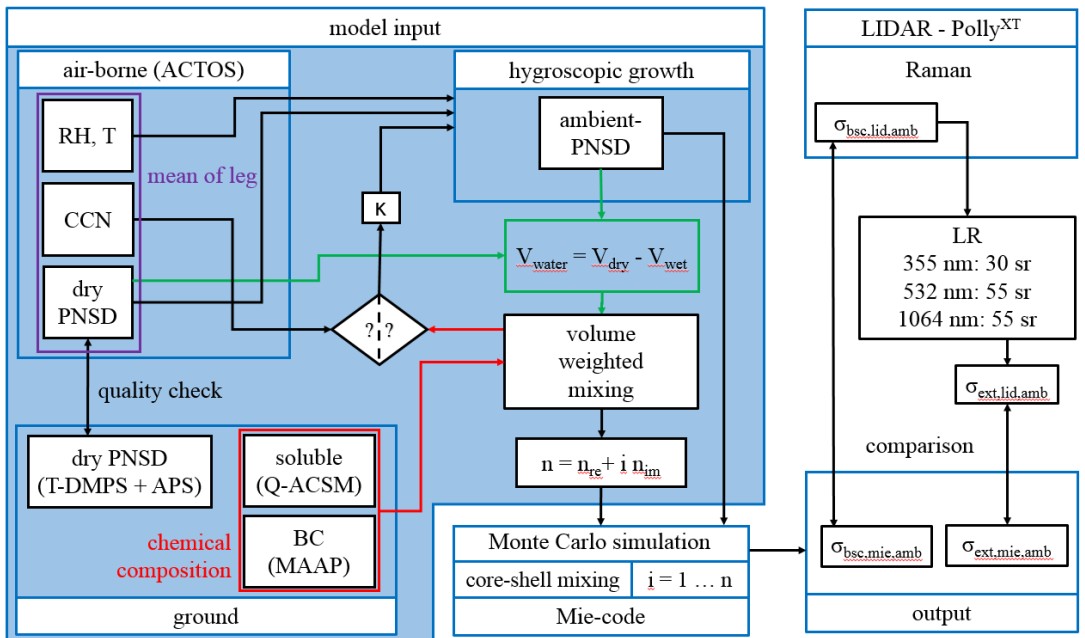

**Figure 7: Flowchart of the algorithm to convert airborne in situ measurements to aerosol particle optical properties in ambient state.**

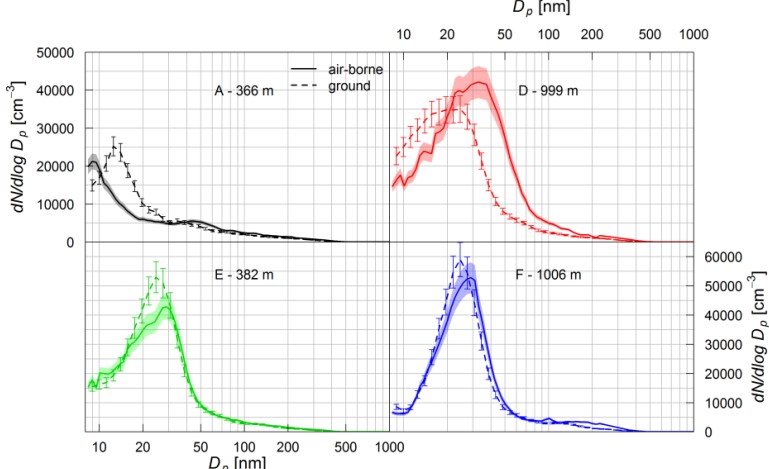

**Figure 8: Airborne (solid lines) mean PNSD at standard conditions recorded during leg A (black), D (red), E (green), and F (blue) with the corresponding PNSD measured at Melpitz. Error-bars and shaded areas represent the Tropos-standard uncertainty of Tropos-built MPSS systems of 10 %.**



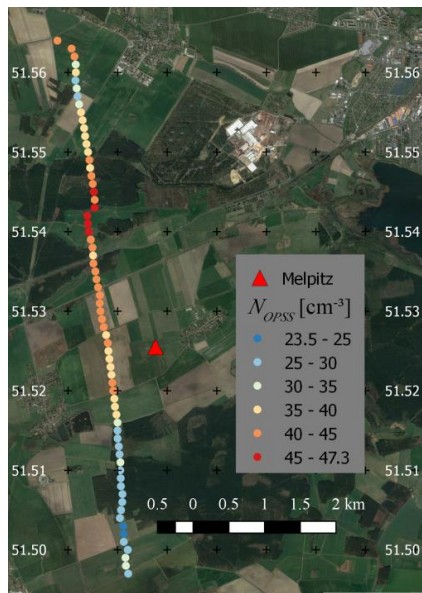

**Figure 9: Horizontal distribution of the mean PNC ($N_{OPSS}$) for particles with an optical diameter of 356 nm to 2.8 µm within layer of 950 to 1050 m above ground recorded between 13.06 and 13.34 UTC on Sept. 14 (leg D and F of flight 14b). The more reddish the symbol the higher is the concentration. The red triangle represents the measurement site in Melpitz.**





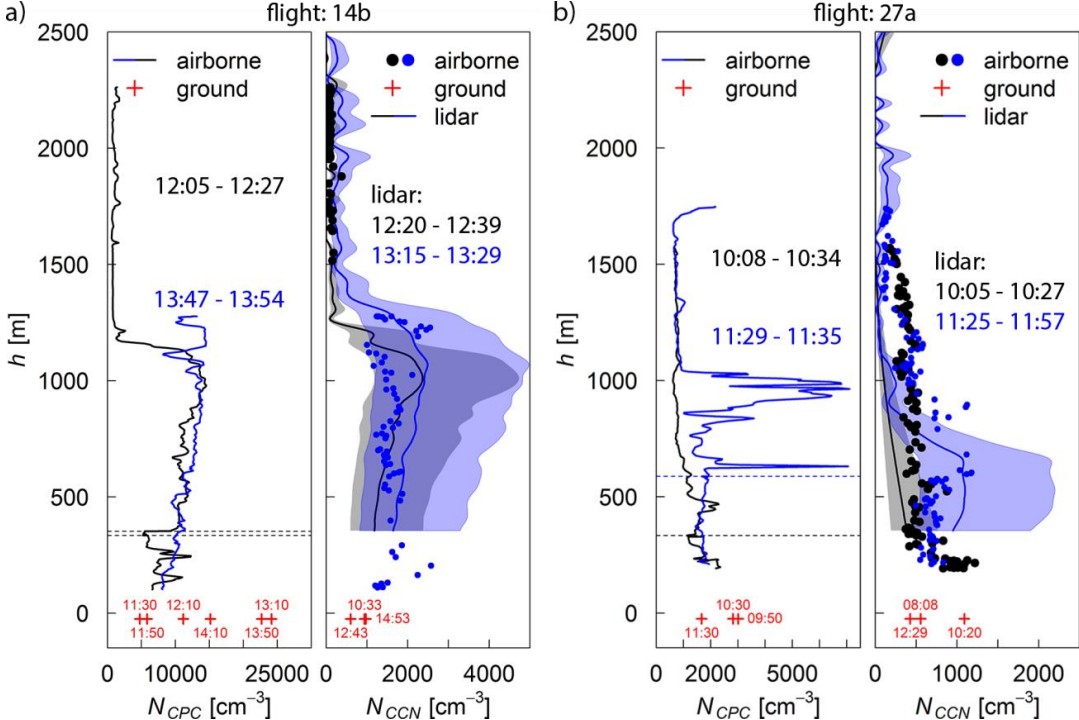

**Figure 10: Both panels show on the left side PNC measurements. Airborne-based profiles of the PNC derived with the CPC on ACTOS ($N_{CPC}$) are shown as solid lines On the right side of each panel, CCN-NC are shown derived with the mCCNc on ACTOS ($N_{CCN,mCCNc}$, dots) for a supersaturation of 0.2 % and derived with the approach of Mamouri and Ansmann (2016) for 0.15 % supersaturation are shown ($N_{CCN,lid}$, solid lines with shaded area). The shaded area represents the given uncertainty of a factor of two of the approach of Mamouri and Ansmann (2016). Red crosses symbolize the ground measurement of the respective parameter at the given time in UTC. In panel a) the two profiles of PNC and CCN-NC were determined during flight 14b in the time period of 12.05 to 12.27 UTC (black) and in the period between 13.47 and 13.54 UTC (blue) are shown. The lidar profiles were averaged between 12.20 and 12.39 UTC (black) and 13.15 and 13.29 UTC (blue). Panel b) shows profiles of the PNC and CCN-NC determined during flight 27a in the time period from 10.08 to 10.34 UTC (black) and from 11.29 to 11.35 UTC (blue). The two CCN-NC profiles derived with lidar measurements ($N_{CCN,lid}$) are shown for the time period between 10.05 and 10.27 UTC (black line) and 11.25 and 11.57 UTC (blue line). The black or blue dashed lines in the panels representing the height in which the respective profiles (black or blue) were put together.**





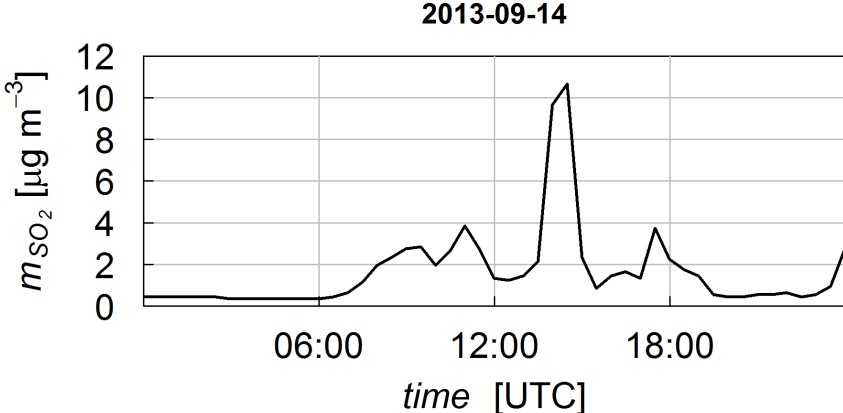

**Figure 11: SO₂ mass concentration measured at Melpitz observatory on Sept. 14, 2013.**

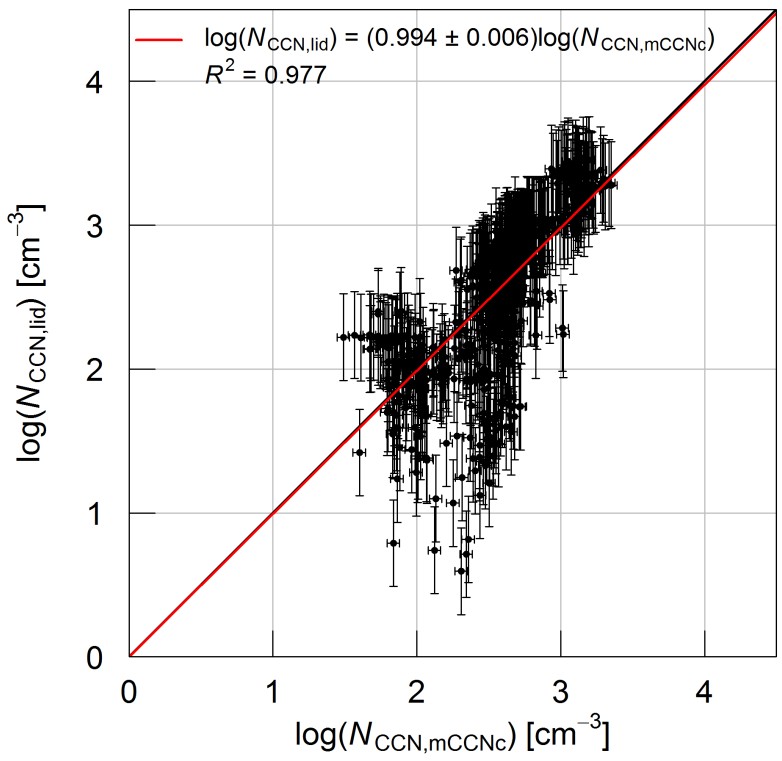

**Figure 12: Correlation of the logarithmized (base 10) CCN-NC derived with the approach of Mamouri and Ansmann (2016) ($N_{CCN,lid}$) and directly measured with the mCCNc on ACTOS ($N_{CCN,mCCNc}$) for six profiles conducted during three flights (14a, 14b and 27a). Each profile has its associated lidar profile. Red line represents the line of fit and the black line the 1:1 line. Error bars represent the uncertainty of the lidar-based approach of a factor 2 and the 10 % uncertainty of the mCCNc on ACTOS.**



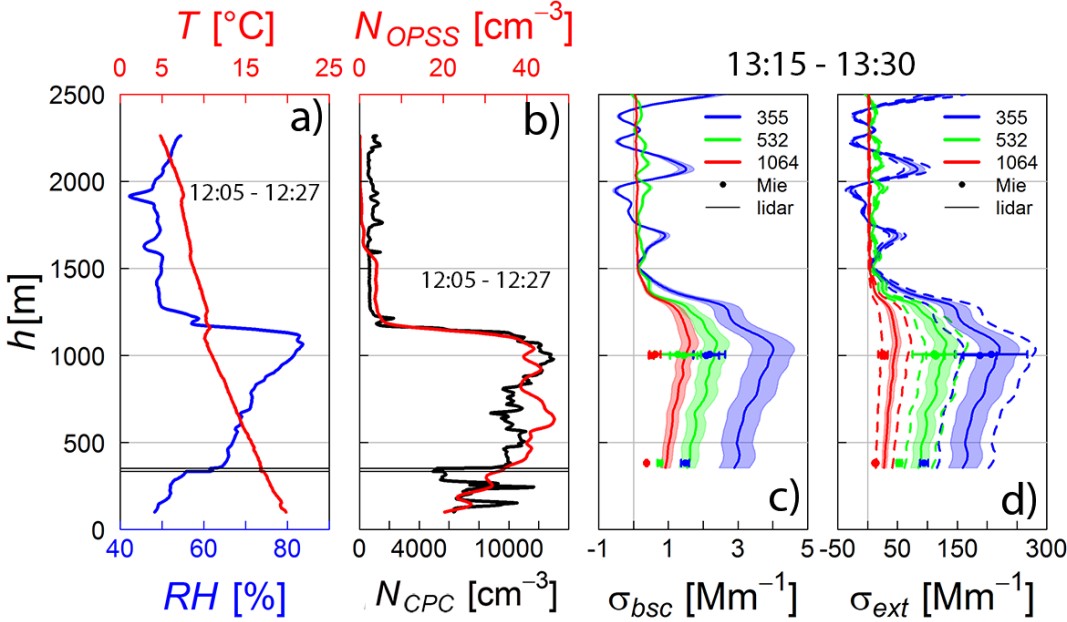

**Figure 13: Panel (a) shows the profiles of the ambient relative humidity (*RH*, blue) and temperature (*T*, red), whereas (b) illustrates the PNC derived with the CPC on ACTOS (*N*CPC, black) and the PNC derived with the OPSS (*N*OPSS, red). The black thin horizontal lines in (a) and (b) mark the height where the profile was composed out of several vertical parts of the flight 14b. The profile was flown between 12.05 and 12.27 UTC in the beginning of the flight. Panel (c) and (d) display the lidar-based particle light backscatter coefficient (σbsc,lid(λ)) and the particle light extinction coefficient (σext,lid(λ)) for three wavelengths (355 (blue), 532 (green) and 1064 nm (red)) averaged over the period 13.15 - 13.30 UTC of Sept. 14. σbsc,lid(λ) and σext,lid(λ) were smoothed (algorithm uses each 6th data point) within 350 and 2500 m height. Also the results of the airborne-based particle light backscatter (σbsc,mie(λ)) and extinction coefficient (σext,mie(λ)) are shown as colored dots for three different wavelengths (coloring same as for the lidar-based values). The error bars of the dots indicate the three times standard deviation of mean value over 50 runs of the Mie-algorithm calculations. The shaded area around the lidar profiles marks the 15% error. The solid lines in the extinction panel (d) represents the profile for the extinction calculated out of the backscattering using the *LR* presented here (55 sr for 355 and 532 nm, 30 sr for 1064 nm). The dashed line signs the extinction profile using a LR ± 15 sr.**



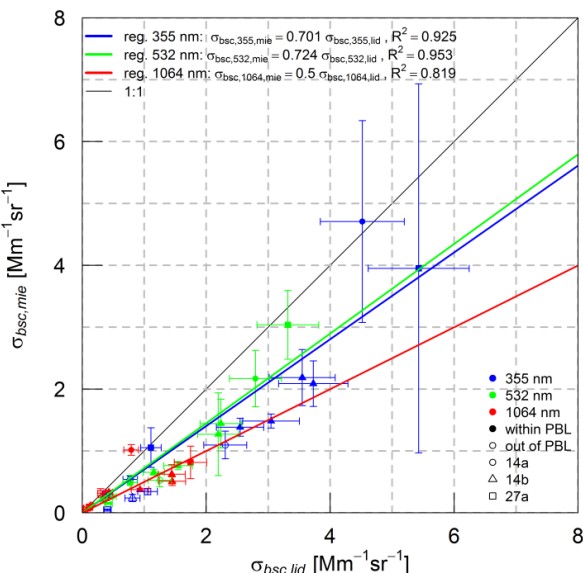

**Figure 14:** Scatterplot of the airborne-based ($\sigma_{bsc,mie}$) and the lidar-based ($\sigma_{bsc,lid}$) particle light backscatter coefficient for all horizontal legs during the investigated days for wavelengths λ = 355 (blue), 532 (green) and 1064 nm (red). The error-bars represent the assumed 15%-error for the lidar-measurements and the three times standard-deviation of mean of the result of the Mie-calculations.

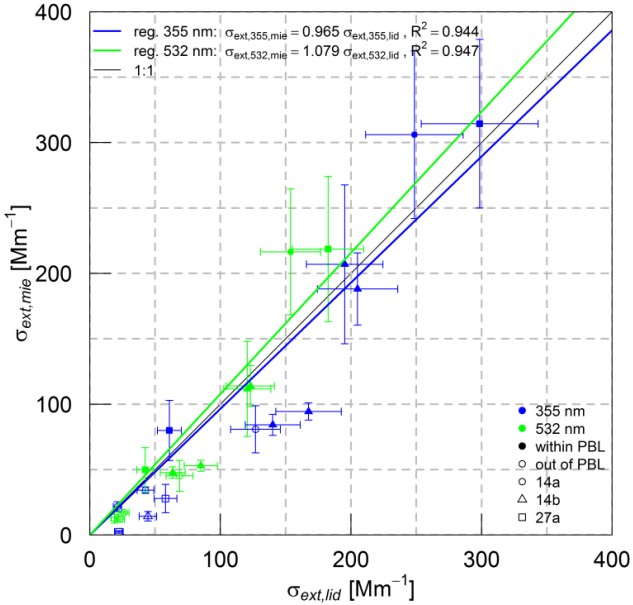

**Figure 15:** Scatterplot of the airborne-based ($\sigma_{ext,mie}$) and the lidar-based ($\sigma_{ext,lid}$) particle light extinction coefficient for all horizontal legs during the investigated days for the wavelengths λ = 355 (blue) and 532 nm (green). $\sigma_{ext,lid}$ derived with a LR of 55 sr. The error-bars represent the assumed 15 % - error for the lidar-measurements and the three times standard-deviation of mean of the result of the Mie-calculations.



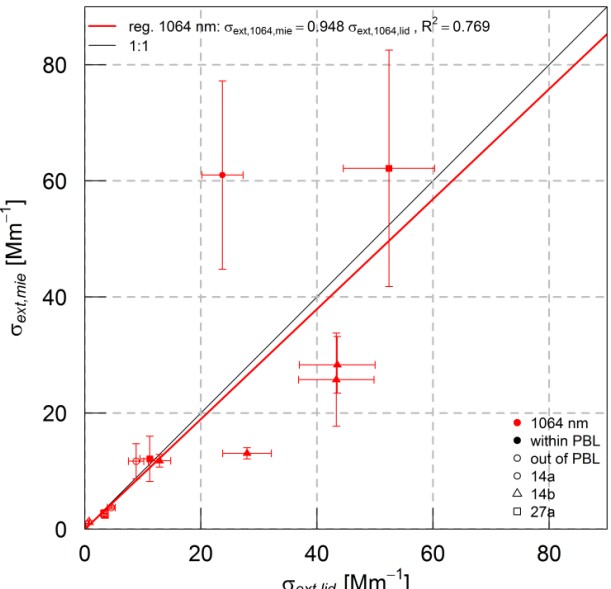

**Figure 16: Scatterplot of particle light extinction coefficient derived with Mie-calculations ($\sigma_{ext,mie}$) and lidar-based ($\sigma_{ext,lid}$) for all horizontal legs during the investigated days for $\lambda = 1064$ nm. $\sigma_{ext,lid}$ derived with a lidar ratio ($LR$) of 30 sr. The error-bars represent the assumed 15%-error for the lidar-measurements and the three times standard-deviation of mean of the result of the Mie-calculations.**

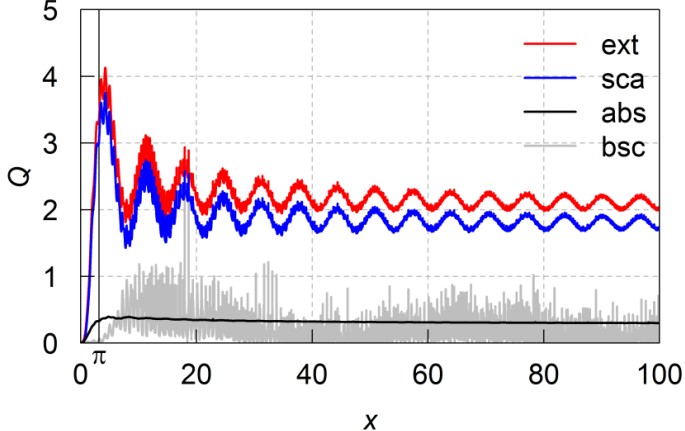

**Figure A1: Mie-based particle light extinction, scattering, absorption and backscatter efficiency ($Q_{ext}$, $Q_{sca}$, $Q_{abs}$, and $Q_{bsc}$) depending on the size parameter $x$ of layered aerosol particles with a core of eBC and a shell for a wavelength of 355 nm. The volume fraction of eBC is 0.05. Thee refractive index of the core and shell taken from Table 3.**