# Peer review of "Helicopter-borne observations of the continental background aerosol in combination with remote sensing and ground-based measurements."

_Atmospheric Chemistry and Physics, 2017_

## Referee Comment (RC1) · Anonymous Referee #1 · 11 Sep 2017

Düsing et al. (2017) describe a closure study using the airborne ACTOS payload. Their goal was to evaluate the extent to which ground-based measurements were representative of vertically-resolved (airborne) measurements. They focused on aerosol optical properties measured or inferred by ground-based lidar, which includes backscatter coefficient, CCN number concentration (CCN-NC), and particle hygroscopicity. Overall, the measurements and analysis presented in this manuscript are of a high quality and I have only a few scientific comments. My major comment would be that the presentation of these results can be significantly improved before publishing.

[Figure]

The conclusions of this manuscript are currently lost in an excess of detail which is presented without clearly signalling the takehome message. As a prime example, the goal of the paper is not stated until the third paragraph of the abstract. As another example, section 4.3.1 titled "case study of flight 14b", begins with a review of basic flight statistics but goes on to perform an evaluation of the lidar backscatter backscatter coefficient data. The latter is clearly the main goal of the case study, and the reader needs to be informed of that by changing to a more descriptive title. Especially for a long paper which addresses multiple topics, it is important to provide a clear manuscript structure. The multiple topics discussed here include lidar, CCN and aerosol optics, which will probably attract readers from a variety of backgrounds who will each want to read only one of those topics. It would be better to summarize the flight statistics in a dedicated results section followed by sections focused on the take home messages, like "In situ versus lidar measurements of Bext", "Vertical profiles of aerosol hygroscopicity", etc. These are just examples.

My other major criticism is that the explanations of some observations are too speculative. On line 20 of page 18, the authors argue that the Mie calculations underestimated the backscatter coefficient because the upper cutoff of the inlet system was 2 microns. When an argument like this is presented, it should be backed up by hard data. For example, add a statement like "Using flight 14b as an example, we calculated that, at 5 microns, only 2 particles per cubic centimeter would be required to close the gap between the Mie calculations and the lidar measurements."

Another example of an incomplete or fragmented argument is on line 1 of page 20. The authors first speculate that aerosol hygroscopicity from the CCNC was influenced by supermicron particles, not measured by the ACSM. A few sentences later, they state that the CCNC hygroscopicity is only valid in the size range of the derived critical diameter. These two statements contradict each other because the critical diameters will be less than a micron.

My recommendation would be that the authors rewrite the entire text to be more focused on the take home messages and minimize speculation. The manuscript would then be greatly improved. However, I do not consider the manuscript unpublishable in its current form.

Other comments

-Page 10, line 15-23. The authors state that "aerosol particles consists of a core surrounded by a shell" gave "the best agreement between modeled and measured hemispheric backscatter coefficients for Melpitz" and cite Ma et al (2014). I was not aware of that conclusion, so I consulted the cited paper. I do not see anything about "best agreement" in Ma et al. In Ma et al, Table 2 suggests that no conclusions could be made about mixing state in their work, and the authors did not make any such conclusions. Mixing state assumptions do not appear to be important at Melpitz.

-Page 17, line 30. It is not correct to delete negative values because they are unphysical. A negative value is only unphysical if it comes along with a confidence interval less than its magnitude. Otherwise, it is the same as a zero. On the other hand, if a negative value is still negative after considering the confidence interval, it means the confidence interval is too small. In this case, it is definitely misleading to delete negative values, which are now telling you that there are fundamental problems in the calculation. If the authors have systematically deleted negative values, this would explain the overall high bias of the lidar results.

-Page 21, line 15. "This study shows that the aerosol type dependent intensive property of the LR" "leads to uncertainties in particle light extinction profiles". This is a strong conclusion that I did not personally see demonstrated in this manuscript.

Page 22, line 4. The variation of Qbsc with x does not prove that a precise calculation of Qbsc (total) requires a very precise determination of the PNSD. The Qbsc (x) will be smoothed out when integrating over a size distribution.

Minor comments

-The abstract is far too long. Although ACP does not enforce abstract length require-ments, this abstract reads more like a thesis abstract than a manuscript abstract, it doesn't communicate the manuscript's conclusions effectively.

-The ACSM measures organics which vaporize at 600 degrees C. This can include water insoluble material such as hydrocarbons emitted by traffic. I am not criticizing the data analysis, only the language.

-On line 18 of page 2, the aerosol-radiation interaction radiative forcing is quoted as -0.35 W/m2 without mentioning the uncertainty range (-0.85 to +0.15). It is the uncer-tainty range and not the value which is important here.

-Page 3 line 18 please provide a citation for the GAW network.

-Page 3 line 34 onwards, this paragraph seems out of place. Are you describing a shortcoming of airborne measurements, or describing the methodology necessary to compare in situ and remote sensing data? Some guiding words are missing.

-Page 4, line 3, please state explicitly which two of the challenges.

-Equation 1 is missing the dynamic shape factor. Please include it. I understand that you assume it is part of your assumed density, but it still needs to be included.

-Page 18, line 7. Why three times the standard deviation? This would imply a confi-dence interval of 99%. There is no point to asserting such a high confidence interval, when modeling assumptions have been significant.

-Page 19, line 24. "mainly the non-observed size range in the PNSD". How do you know this is the main cause?

-Page 21, line 8, "complex behaviour" is not a satisfying explanation.

---

## Referee Comment (RC2) · Anonymous Referee #2 · 27 Sep 2017

**GENERAL COMMENTS**

This paper presents an interesting study combining in-situ and remote sensing measurements of the atmospheric aerosol. On the basis of ground-based and airborne measurements, the authors investigate the representativeness of ground-based aerosol microphysical properties for the boundary layer. Furthermore, the CCN retrieved from remote sensing are compared with airborne in-situ measurements. Finally, a closure study is performed between the optical properties derived from lidar measurements and those derived from Mie theory using the airborne measurements.

In this way, the authors do a rather complete exploitation of the available data set.

The paper is appropriate for Atmospheric Chemistry and Physics journal.

Nevertheless, some aspects must be improved before the paper would be accepted for publication. Thus, although the results are interesting and significative the authors must improve the presentation of results. In this sense, the English wording and the structure of the paper require some attention. Particularly, a paper covering such variety of topics requires a description of the structure at the end of the introduction and an appropriate transition from one section to the following. Furthermore, some parts of the discussion require clarification.

PARTICULAR COMMENTS The authors must provide some comments on the challenge of using a simple Lidar Ratio for the whole atmospheric column, especially when some layering is evident in the aerosol profile. The impact of this assumption is really relevant for the extinction profiles derived from the lidar measurements, that in a last stage are used for deriving the CCN profiles.

Related to the previous comment, the statement "we consider an uncertainty in the lidar measurements of up to 15 %" must be clarified.

It is necessary that the authors provide details on the determination of the Planetary Boundary Layer Height presented in Fig 10.

The choice of colors, for the different variables shown, in Fig 10 must be improved for the sake of readiness and to facilitate the understanding of the discussion.

It would be worthy to include error bars, associated to the uncertainty of the measurements of the different variable, as the authors do for the case of lidar derived CCN.

Concerning the uncertainties of lidar derived CCN the authors must explain what they are stating as "factor two".

Considering the discrepancy on CCN measured and derived from lidar it would be

worthy to explore the use of different approaches for the lidar derived CCN, specially considering the features of the aerosol in the layers showing the largest discrepancy.

The discussion on the correlation between mCCN and lidar derived CCN must be corrected, as far as the logarithm of any variable have not units, the correct consideration would be to divide the CCN by 1 cm-3 and then compute the logarithm of a unitless variable. The axis in Fig 12 must be appropriately corrected. So the right comment must be: "On the second glance, in Figure 12 its clearly visible, that the lidar approach overestimates the airborne CCN-NC measurements for values of log10(NCCN,mCCNc) from 2.7 to 3.4 (NNCN, mCCNc in the ranges 500 to 2500 cm-3) by a factor of two, whereas in the range from log10(NCCN,mCCNc) = 1.8 to 2.5 (NCCN, mCCNc 60 to  $\sim$ 320 cm-3) the lidar approach underestimates. This indicates different aerosol types and explains the low correlation. Note, in the regime up to log10(NCCN,mCCNc) = 1.8 the lidar approach.."

Considering the procedure followed in the derivation of the extinction profile from the backscattering profile using a fixed lidar ratio the comment: "The lidar profiles of  $\sigma$ bsc and  $\sigma$ ext show the same behavior" is rather obvious and can be omitted.

In Tables B1 and B2 and in the discussion on LR just indicate the uncertainty with nor more than two significant figures (one if the most significant one is larger than 3) and then express the value plus minus uncertainty appropriately. For example:

68.84±10.33 must be 69+-10

23.21±3.48 must be 23+-3

153.75±23.06 must be 153+-23

Just correct the small number of typos along the text.

The quality of some figures must be improved, specially the labels of Figures 14 -16.

СЗ

2017.

---

## Author Comment (AC1) · 8 Nov 2017

Response to Referee #1

*First of all, we appreciate the detailed review of the referee and we acknowledge the effort he/she spend for this. The response to the reviewer is given below answering the remarks point by point.*

Düsing et al. (2017) describe a closure study using the airborne ACTOS payload. Their goal was to evaluate the extent to which ground-based measurements were representative of vertically-resolved (airborne) measurements. They focused on aerosol optical properties measured or inferred by ground-based lidar, which includes backscatter coefficient, CCN number concentration (CCN-NC), and particle hygroscopicity. Overall, the measurements and analysis presented in this manuscript are of a high quality and I have only a few scientific comments. My major comment would be that the presentation of these results can be significantly improved before publishing.

The conclusions of this manuscript are currently lost in an excess of detail which is presented without clearly signalling the takehome message. As a prime example, the goal of the paper is not stated until the third paragraph of the abstract. As another example, section 4.3.1 titled "case study of flight 14b", begins with a review of basic flight statistics but goes on to perform an evaluation of the lidar backscatter backscatter coefficient data. The latter is clearly the main goal of the case study, and the reader needs to be informed of that by changing to a more descriptive title. Especially for a long paper which addresses multiple topics, it is important to provide a clear manuscript structure. The multiple topics discussed here include lidar, CCN and aerosol optics, which will probably attract readers from a variety of backgrounds who will each want to read only one of those topics. It would be better to summarize the flight statistics in a dedicated results section followed by sections focused on the take home messages, like "In situ versus lidar measurements of Bext", "Vertical profiles of aerosol hygroscopicity", etc. These are just examples.

*Author response: We updated the abstract. In particular we mentioned the goal of the paper in the first paragraph now. In addition, we slightly shortened the abstract, but due to the variety of topics in this paper, a further shortening would skip content of the paper. The new abstract follows below and the red text indicates new text:"* **This paper examines the representativeness of ground-based in-situ measurements for the planetary boundary layer (PBL) and conducts a closure study between airborne in-situ and ground based lidar measurements up to an altitude of 2300 m. The related measurements were carried out in a field campaign within the framework of the High-Definition Clouds and Precipitation for Advancing Climate Prediction (HD (CP)²) Observational Prototype Experiment (HOPE) in September 2013 in a rural background area of Central Europe.**

**The helicopter-borne probe ACTOS (Airborne Cloud and Turbulence Observation System) provided measurements of the aerosol particle number size distribution (PNSD), the aerosol particle number concentration (PNC), the number concentration of cloud condensation nuclei (CCN-NC) and meteorological atmospheric parameters (e.g. temperature and relative humidity). These measurements were supported by the ground-based 3+2 wavelength polarization lidar system Polly$^{XT}$, which provided profiles of the particle backscatter coefficient ($\sigma_{bsc}$) for three wavelengths (355, 532 and 1064 nm). Particle extinction coefficient ($\sigma_{ext}$) profiles were obtained by using a fixed backscatter-to-extinction ratio (also lidar ratio, LR). A new approach was used to determine profiles of CCN-NC for continental aerosol. The results of this new approach were consistent with the airborne in-situ measurements within the uncertainties.**

**In terms of representativeness,** **the PNSD measurements on ground showed a good agreement with the measurements provided with ACTOS for lower altitudes. The ground-based measurements of PNC and CCN-NC are representative for the PBL when the PBL is well mixed. Locally isolated new particle formation events on ground or at the top of the PBL led to vertical variability in the here presented cases and ground-based measurements are not entirely representative for the PBL.**

**Based on Mie-theory, optical aerosol properties under ambient conditions for different altitudes were determined using the airborne in-situ measurements and were compared with the lidar measurements.**

*The investigation of the optical properties shows that on average the airborne-based particle light backscatter coefficient is for 1064 nm 50.1 % smaller than the measurements of the lidar system, 27.4 % smaller for 532 nm and 29.5 % smaller for 355 nm. These results are quite promising, since in-situ measurement based Mie-calculations of the particle light backscattering are scarce and the modelling is quite challenging. In contradiction, for the particle light extinction coefficient we found a good agreement. The airborne-based particle light extinction coefficient was just 8.2 % larger for 532 nm and 3 % smaller for 355 nm, for an assumed lidar ratio (LR) of 55 sr. The particle light extinction coefficient for 1064 nm was derived with a LR of 30 sr. For this wavelength, the airborne-based particle light extinction coefficient is 5.2 % smaller than the lidar-measurements. For the first time, the lidar ratio of 30 sr for 1064 nm was determined on the basis of in-situ measurements and the LR of 55 sr for 355 and 532 nm wavelength was reproduced for European continental aerosol on the basis of this comparison."*

*We rearranged some passages of the manuscript. Especially Sect. 4.3.1, 4.3.2 and 4.3.3 were changed. The new title of Sect. 4.3.1 is now "Vertical structure during the flights" and explains with the example of a vertical profile from flight 14b the general vertical structure during the measurement flights. We included in Sect. 4.3.1 the following red marked text: "Using the example of a profile from flight 14b, we first illustrate the vertical structure of the atmosphere in the investigated area. Fig. 13 shows the vertical structure of in-situ measured RH and T in panel a), in panel b) the aerosol particle number concentration measured by the CPC ($N_{CPC}$) on ACTOS and by integrating the OPSS number size distribution $N_{OPSS}$. Furthermore, the vertical profiles of the particle backscatter coefficient derived with the lidar ($\sigma_{bsc,lid}$; colored solid lines) and with the Mie-calculations calculated for leg D, E and F ($\sigma_{bsc,mie}$; colored dots with error bars) for the three lidar wavelengths is shown in panel c). Additionally, profiles of the particle extinction coefficient are shown for both lidar derived and Mie-based in panel d). The shaded area around the lidar profiles indicates the assumed 15 % uncertainty and the dashed lines with the respective colors around the lidar-based extinction profiles indicate the calculated by using the particle extinction coefficient profile derived of the particle backscatter coefficient profiles with LR ± 15 sr larger and smaller, respectively. In this example, the profiles of T and RH show an inversion at approximately 1200 m." and furthermore "During this campaign, we found a qualitatively similar structure also in the other investigated flights but the quantity of the shown parameters, for instance the height of the PBL and the PNC within the PBL was different." For Sect. 4.3.2 and 4.3.3 we now discuss the optical closure with an example and also we give an overview of all investigated data-points.*

My other major criticism is that the explanations of some observations are too speculative. On line 20 of page 18, the authors argue that the Mie calculations underestimated the backscatter coefficient because the upper cutoff of the inlet system was 2 microns. When an argument like this is presented, it should be backed up by hard data. For example, add a statement like "Using flight 14b as an example, we calculated that, at 5 microns, only 2 particles per cubic centimeter would be required to close the gap between the Mie calculations and the lidar measurements."

*Author response: We did some calculations for a monodisperse aerosol with two median mode diameter of 2 and 5 μm with two aerosol particles in total. We extended Sect. 4.2.2 with the following: For example, we have calculated the particle backscatter for a monodisperse aerosol at a wavelength of 355,532 and 1064 nm. The same conditions applied here as in the horizontal leg F of flight 14b. The monodisperse size distribution was created with Eq. (11):*

$$\frac{dN}{d\log D_{\mathrm{p}}}(D_{\mathrm{p}}) = \frac{N}{\sqrt{2\pi}\,\log_{10}(\sigma)} \exp\left[-\frac{1}{2}\left(\frac{\log_{10}\left(\frac{D_{\mathrm{p}}}{\overline{D_{\mathrm{p}}}}\right)}{\log_{10}(\sigma)}\right)\right], \qquad (11)$$

*where N denotes the total particle number concentration in the mode, σ the geometric standard deviation, and $\overline{D_p}$ the median diameter of the mode. With a volume fraction of 0.037 of eBC, a N of 2 cm$^{-3}$, a σ of 1.1*

*and $\overline{D_p}$ of 2 μm would cause a particle backscattering of 1.44 (355 nm), 2.29 (532 nm) and 1.7 Mm$^{-1}$ sr$^{-1}$ (1064 nm). For monodisperse aerosol with a median diameter of 5 μm the calculation result in 5.39 (355 nm), 2.09 (532 nm), and 9.09 Mm$^{-1}$ sr$^{-1}$ (1064 nm). This configuration is more than enough to close the gap between the calculations and the observations.*

Another example of an incomplete or fragmented argument is on line 1 of page 20. The authors first speculate that aerosol hygroscopicity from the CCNC was influenced by supermicron particles, not measured by the ACSM. A few sentences later, they state that the CCNC hygroscopicity is only valid in the size range of the derived critical diameter. These two statements contradict each other because the critical diameters will be less than a micron.

*Author response: We considered that the section was contradicted. Therefore we were more focused on the measurements of the Q-ACSM now. For this purpose we cited Martin et al. (2011). They showed a hygroscopicity closure with basically the same set-up as used as in this paper. Although they considered Arctic Summer aerosol we think their conclusions can support our thoughts. The differences in the hygroscopicity derived with both methods (CCNc and Q-ACSM) are explained more consistently in our opinion now. The manuscript contains now the red marked text: "Furthermore, the hygroscopicity determined by the CCNc is only valid in the size range of the critical diameter. Calculations of the aerosol optical properties under ambient conditions may therefore not be as representative as calculations with hygroscopicity from the Q-ACSM measurements.*

*Based on the ground-based CCN-NC measurements, the hygroscopicity of the aerosol particles was also derived. The resulting kappa from both methods is shown in Table 5. For the two days considered in this study, CCNc measurements on the ground led to lower values than the Q-ACSM measurements. Similar results were also observed by Martin et al. (2011) in the case of Arctic Summer aerosol. They predicted on the basis of AMS (Aerosol Mass Spectrometer) measurements consistently higher CCN-NC (correlated to hygroscopicity) than were measured with a CCNC for various supersaturations.*

*Organics could lead to an overestimation of the Q-ACSM based hygroscopicity. Martin et al. (2011) obtained the best agreements if they regarded the organic substances as almost insoluble in water, which could indicate that in our case either the water-insoluble material was not detected or the detected organic substances had a lower hygroscopicity.*

*In addition, both measurements may differ, since the Q-ACSM detects the aerosol in its completeness (PM1), while as mentioned above, the hygroscopicity of CCNc measurements is only valid for the critical diameter range."*

*Cited Paper: Martin, M., Chang, R. Y.-W., Sierau, B., Sjogren, S., Swietlicki, E., Abbatt, J. P. D., Leck, C., and Lohmann, U.: Cloud condensation nuclei closure study on summer arctic aerosol, Atmos. Chem. Phys., 11, 11335-11350, https://doi.org/10.5194/acp-11-11335-2011, 2011.*

My recommendation would be that the authors rewrite the entire text to be more focused on the take home messages and minimize speculation. The manuscript would then be greatly improved. However, I do not consider the manuscript unpublishable in its current form.

Other comments

- Page 10, line 15-23. The authors state that "aerosol particles consists of a core surrounded by a shell" gave "the best agreement between modeled and measured hemispheric backscatter coefficients for Melpitz" and cite Ma et al (2014). I was not aware of that conclusion, so I consulted the cited paper. I do not see anything about "best agreement" in Ma et al. In Ma et al, Table 2 suggests that no conclusions could be made about mixing state in their work, and the authors did not make any such conclusions. Mixing state assumptions do not appear to be important at Melpitz.

  *Author response: Table 2 in Ma et al., 2014 may not provide significant differences between the different mixing state assumptions. Though, for the hemispheric backscattering the slope of the correlation between measurements and calculations is closest to unity. Furthermore, Ma et al., 2014 cites several paper:* **"These conceptual models have been widely used to assess aerosol optical properties and direct radiative forcing. Among the internal mixture models, the core–shell mixture model – suggesting that light-absorbing carbon cores are surrounded by shells of less absorbing components, has been shown to yield more realistic results than the homogeneous internal mixture model (Jacobson, 2001; Chandra et al., 2004; Katrinak et al., 1992, 1993; Ma et al., 2012)."** *showing that the core-shell mixture approach leads to more realistic results.*

- Page 17, line 30. It is not correct to delete negative values because they are unphysical. A negative value is only unphysical if it comes along with a confidence interval less than its magnitude. Otherwise, it is the same as a zero. On the other hand, if a negative value is still negative after considering the confidence interval, it means the confidence interval is too small. In this case, it is definitely misleading to delete negative values, which are now telling you that there are fundamental problems in the calculation. If the authors have systematically deleted negative values, this would explain the overall high bias of the lidar results."

  *Author response: You are right and in fact we have not deleted negative values in the profiles as this would certainly bias the results. What we mean is that in height regions at which the signal-to-noise ratio of the lidar is too low we cannot determine the optical aerosol properties with sufficient accuracy. This is mostly the case when the aerosol concentration is very low. In this case, measurement errors are too high and thus were left out for comparison because the noise led to negative measurements. We therefore removed most of the cases above the planetary boundary layer since in this cases the lidar signal was too low. We updated the tables B1, B2 and 6, and also Fig. 15 to 16. But, since the model also calculated low particle extinction and backscatter coefficients (close to zero in the cases with low aerosol load), the slope and R² of the correlation did not changed significantly. For example: old slope of backscatter coefficient correlation for 1064 nm is 0.5 (R² = 0.819), new is also 0.499 (R² = 0.819). For 355 nm: old is 0.701 (R² = 0.925) and new one is 0.705 (R² = 0.928).*

  *We have changed the text accordingly to:* *"For fields marked with "-", the signal-to-noise ratio of the lidar within the respective height region was too low to retrieve aerosol properties with high accuracy and therefore were not used for comparison."*

  *The updated figures you can find below (Fig. R1, R2, and R3). Also they show the new correlation.*

  Page 21, line 15. "This study shows that the aerosol type dependent intensive property of the LR" "leads to uncertainties in particle light extinction profiles". This is a strong conclusion that I did not personally see demonstrated in this manuscript.

*Author response: It is true that the explanations in the paper do not provide any information about the influence of the lidar ratio on extinction. Therefore, we have removed the above sentence. However, model calculations in the different heights showed different lidar ratios (see Sect. 4.3.3), so that the assumption of a height-independent lidar ratio can lead to deviations in the aerosol optical depth.*

- Page 22, line 4. The variation of $Q_{bsc}$ with x does not prove that a precise calculation of $Q_{bsc}$ (total) requires a very precise determination of the PNSD. The $Q_{bsc}(x)$ will be smoothed out when integrating over a size distribution.

*Author response: The reviewer is right. We have checked the statement with a modelled mono-disperse aerosol particle size distribution. For this we first use a PNSD with 14 size-bins in the size-range covered by the used OPSS and calculated the particle backscattering. We then calculated the particle backscattering for the same PNSD with 50 size-bins. The differences between these approaches were marginal. Therefore, we have removed the complex behavior of the backscattering effect as a reason and changed the statement that the particles that are most effective for backscattering have hardly been detected, as they are larger than those that can be detected by the OPSS. The new section is as follows: "Furthermore, comparison of Mie-theory-based and lidar-based particle light backscatter coefficients implies that the here used setup cannot provide a complete data-base to reproduce the "real" particle light backscatter coefficient since the investigated size-range seems to be too small. This can be explained by the behavior of the backscatter efficiency of aerosol particles in the narrow scattering angle window in 180° direction (cf. Fig. A1; high backscatter efficiency of particles ~6 times larger in diameter than the incoming radiation)."*

Minor comments

- The abstract is far too long. Although ACP does not enforce abstract length requirements, this abstract reads more like a thesis abstract than a manuscript abstract, it doesn't communicate the manuscript's conclusions effectively.

*Author response: We agree that the abstract was too long. We rewrote the abstract but since the variety of topics it is not much shorter than before (see author response above; red part is new). We think the conclusion is communicated now.*

The ACSM measures organics which vaporize at 600 degrees C. This can include water insoluble material such as hydrocarbons emitted by traffic. I am not criticizing the data analysis, only the language.

*Author response: We changed the section describing the Q-ACSM as follows (red marked parts): "In this study, a data-set of the continuously running Quadrupole Aerosol Chemical Speciation Monitor (Q-ACSM, Aerodyne Res. Inc, ARI, Billerica, MA.; Ng et al., 2011) was used. The Q-ACSM detects non-refractory particulate matter in the fine regime (NR-PM1) that vaporizes at around 600 °C with a time resolution of about 25 minutes. The included mass spectrometer separates the vaporized material into $SO_4^{-2}$, $NO^{-3}$, $NH^{+4}$ and organics (Ng et al., 2011). A detailed description of the instrument is provided in Ng et al. (2011) and Fröhlich et al. (2015).*

*Based on these ion measurements, the chemical composition of the aerosol particles itself was derived by a simple ion pairing scheme published by Gysel et al. (2007). Although the measurements can be influenced by water-insoluble hydrocarbons, we consider the species of the*

*aerosol compounds derived with the Q-ACSM to be water-soluble*, since Crippa et al. (2014) has found that all over in Europe, the mass fraction of hydrocarbons in organic compounds is 11 ± 6%. *The major mass fraction of water-soluble compound are in PM$_1$ and are thus also representative for PM$_{2.5}$."*

*We argue, that the measurements of the Q-ACMS can be influenced by hydrocarbon-like substances emitted by traffic. But since the mass-fraction of these compounds is 11 ± 6 % in the detected organics over all Europe (as reported by Crippa et al., 2014) we consider that all compounds (including the organics) to be water-soluble.*

*Paper: Crippa, M., Canonaco, F., Lanz, V. A., Äijälä, M., Allan, J. D., Carbone, S., Capes, G., Ceburnis, D., Dall'Osto, M., Day, D. A., DeCarlo, P. F., Ehn, M., Eriksson, A., Freney, E., Hildebrandt Ruiz, L., Hillamo, R., Jimenez, J. L., Junninen, H., Kiendler-Scharr, A., Kortelainen, A.-M., Kulmala, M., Laaksonen, A., Mensah, A. A., Mohr, C., Nemitz, E., O'Dowd, C., Ovadnevaite, J., Pandis, S. N., Petäjä, T., Poulain, L., Saarikoski, S., Sellegri, K., Swietlicki, E., Tiitta, P., Worsnop, D. R., Baltensperger, U., and Prévôt, A. S. H.: Organic aerosol components derived from 25 AMS data sets across Europe using a consistent ME-2 based source apportionment approach, Atmos. Chem. Phys., 14, 6159-6176, https://doi.org/10.5194/acp-14-6159-2014, 2014.*

- On line 18 of page 2, the aerosol-radiation interaction radiative forcing is quoted as -0.35 W/m2 without mentioning the uncertainty range (-0.85 to +0.15). It is the uncertainty range and not the value which is important here.

  *Author response: Thanks for the response and you are right, the uncertainty is the most important part. Therefore we included the following sentence:* The estimate of the radiative forcing by aerosol-radiation interaction of -0.35 W m$^{-2}$ is very uncertain within the borders of -0.85 to 0.15 W m$^{-2}$ (IPCC, 2013).

- Page 3 line 18 please provide a citation for the GAW network.

  *Author response: We included at the end of the sentence:*
  "(http://www.wmo.int/pages/prog/arep/gaw/measurements.html)".

- Page 3 line 34 onwards, this paragraph seems out of place. Are you describing a shortcoming of airborne measurements, or describing the methodology necessary to compare in situ and remote sensing data? Some guiding words are missing.

  *Author response: We have now clarified the paragraph. We wanted to show that it is disadvantageous that in-situ measurements can change the moisture state of the aerosol particles and that the aerosol is therefore often dried beforehand to obtain reproducible conditions. However, in order to achieve comparability with lidar measurements, the hygroscopic growth of the aerosol particles must be simulated. The approach of the hygroscopicity parameter of Petters et al., (2007) is helpful for this purpose. We changed the paragraph as follows:* "Disadvantageously, both, airborne and ground-based in-situ measurements alter the humidity state of the aerosol (Tsekeri et al., 2017). Therefore, the aerosol is often dried before the particle properties are characterized. A comparability with lidar measurements can be achieved by simulating the ambient condition (e.g. size) of the particles. The hygroscopic properties of the particles which can either be measured or calculated are relevant in this context. The parameterization by Petters et al. (2007) is for this purpose a useful approach to ascertain the hygroscopic growth of the aerosol particles on the basis of their hygroscopicity parameter (κ)."

- Page 4, line 3, please state explicitly which two of the challenges.

  *Author response: To increase the understanding of the section we explicitly mentioned the challenges now via adding the red marked parts in the following: "Within the scope of this article, two of the above-mentioned challenges are addressed by means of sophisticated closure studies: a) ground-based in-situ observations to airborne in-situ observations to investigate the representativeness of ground-based in-situ measurements for the planetary boundary layer, and b) airborne in-situ observations to ground-based remote sensing to cross-check assumptions made in lidar remote sensing."*

- Equation 1 is missing the dynamic shape factor. Please include it. I understand that you assume it is part of your assumed density, but it still needs to be included.

  *Author response: We included the red marked parts in the paper.*

  $$D_{\mathrm{em}} = D_{\mathrm{a}}\sqrt{\frac{\chi\,\rho_0}{\rho_a}}, \tag{1}$$

  *according to DeCarlo et al. (2004), whereby the aerosol particle density is assigned by $\rho_a$ and $\rho_0$ is the standard density of 1.0 g cm$^{-3}$. The dynamic shape factor is represented by $\chi$. In this study we assumed an effective aerosol particle density of 1.6 g cm$^{-3}$, according to Ma et al. (2014) for the fine-mode aerosol. The effective density combines the particle density and dynamic shape factor.*

- Page 18, line 7. Why three times the standard deviation? This would imply a confidence interval of 99%. There is no point to asserting such a high confidence interval, when modeling assumptions have been significant.

- *Author response: In Ma et al., 2014 they also used a 3-times standard deviation as a closure criteria. Since this study uses the basically the same assumptions we decided to use a three times standard deviation as the closure criteria.*

- Page 19, line 24. "mainly the non-observed size range in the PNSD". How do you know this is the main cause?

- *Author response: You are right, because we are not sure that the undetected size-range is the main reason for the large fluctuations in the critical diameter. That's why we replaced "but mainly" with "and also" in the manuscript. Nevertheless, the inaccurate linear approximation of the non-observed size-range leads to deviations in the hygroscopicity parameter when determining the critical diameter, because it reacts very sensitively to small changes in the critical diameter. This can therefore be a main reason for the high variations. However, we have not tested this and it is speculative.*

- Page 21, line 8, "complex behaviour" is not a satisfying explanation.

  *Author response: We removed the word "complex".*

Response to Referee #2

GENERAL COMMENTS

This paper presents an interesting study combining in-situ and remote sensing measurements of the atmospheric aerosol. On the basis of ground-based and airborne measurements, the authors investigate the representativeness of ground-based aerosol microphysical properties for the boundary layer. Furthermore, the CCN retrieved from remote sensing are compared with airborne in-situ measurements. Finally, a closure study is performed between the optical properties derived from lidar measurements and those derived from Mie theory using the airborne measurements. In this way, the authors do a rather complete exploitation of the available data set. The paper is appropriate for Atmospheric Chemistry and Physics journal.

*Author response: We highly appreciate the review and thank the reviewer for the spent effort.*

Nevertheless, some aspects must be improved before the paper would be accepted for publication. Thus, although the results are interesting and significative the authors must improve the presentation of results. In this sense, the English wording and the structure of the paper require some attention. Particularly, a paper covering such variety of topics requires a description of the structure at the end of the introduction and an appropriate transition from one section to the following. Furthermore, some parts of the discussion require clarification.

*Author response: A guideline for the paper is given in the end of the introduction now: "The results of this work are presented as follows. Section 2 describes the experiment with all instruments used. In doing so, we will deal separately with the ground and airborne measurements. A description of the meteorological conditions on the measurement days and an explanation of the algorithm for determining the optical properties of the aerosol under ambient conditions is described in Sect. 3. Section 4 uses case studies to clarify the representativeness of ground-based measurements for the planetary boundary layer. Furthermore, a closure between lidar measurements and airborne measurements is shown. Optical and microphysical aerosol properties (CCN) are discussed. Finally, the results are summarized in section 5."*

PARTICULAR COMMENTS

The authors must provide some comments on the challenge of using a simple Lidar Ratio for the whole atmospheric column, especially when some layering is evident in the aerosol profile. The impact of this assumption is really relevant for the extinction profiles derived from the lidar measurements that in a last stage are used for deriving the CCN profiles.

*Author response: We are thankful for this recommendation. We included the following in the end of section 2.3: "CCN-NC profiles are obtained from particle extinction profiles. These are calculated in this study on the basis of a height constant LR from the particle backscattering coefficients. This assumption cannot represent any possible layers with different aerosol types, as different aerosols differ in LR. The assumption of a constant LR would underestimate or overestimate the particle extinction coefficient compared to an aerosol with a higher or lower LR and thus also the CCN number concentration. "*

Related to the previous comment, the statement "we consider an uncertainty in the lidar measurements of up to 15 %" must be clarified.

*Author response: The following was included in the manuscript to support the assumed 15 % uncertainty: Overall, we consider an uncertainty in the lidar measurements of up to 15 %. Wandinger et al., 2016*

*provides an intercomparison campaign of different EARLINET (European Aerosol Research LIdar NETwork, https://www.earlinet.org/index.php?id=earlinet_homepage) instruments, including the lidar system used in this work (Polly^XT). All shown instruments in Wandinger et al., 2016 had a relative deviation of maximum 10 to 20 % to a reference in both, extinction and backscattering. Polly^XT (le02 in Wandinger et al., 2016) had maximum deviation of less than 10%. Taking into account the uncertainty increase due to the assumed lidar ratio and the shorter average windows we consider 15 % as a maximum uncertainty as appropriate even though we are well aware that the uncertainty is usually lower.*

*Paper: Wandinger, U., Freudenthaler, V., Baars, H., Amodeo, A., Engelmann, R., Mattis, I., Groß, S., Pappalardo, G., Giunta, A., D'Amico, G., Chaikovsky, A., Osipenko, F., Slesar, A., Nicolae, D., Belegante, L., Talianu, C., Serikov, I., Linné, H., Jansen, F., Apituley, A., Wilson, K. M., de Graaf, M., Trickl, T., Giehl, H., Adam, M., Comerón, A., Muñoz-Porcar, C., Rocadenbosch, F., Sicard, M., Tomás, S., Lange, D., Kumar, D., Pujadas, M., Molero, F., Fernández, A. J., Alados-Arboledas, L., Bravo-Aranda, J. A., Navas-Guzmán, F., Guerrero-Rascado, J. L., Granados-Muñoz, M. J., Preißler, J., Wagner, F., Gausa, M., Grigorov, I., Stoyanov, D., Iarlori, M., Rizi, V., Spinelli, N., Boselli, A., Wang, X., Lo Feudo, T., Perrone, M. R., De Tomasi, F., and Burlizzi, P.: EARLINET instrument intercomparison campaigns: overview on strategy and results, Atmos. Meas. Tech., 9, 1001-1023, https://doi.org/10.5194/amt-9-1001-2016, 2016.*

It is necessary that the authors provide details on the determination of the Planetary Boundary Layer Height presented in Fig 10. The choice of colors, for the different variables shown, in Fig 10 must be improved for the sake of readiness and to facilitate the understanding of the discussion.

*Author response: Maybe the description was somewhat misleading. The dashed lines in the figure do not represent the height of the planetary boundary layer. Moreover they symbolize the height in which different vertically flown parts of the flight were joined together to a vertical profile. In the end of the figure caption we added the following sentence.”* **The horizontal black and blue dashed lines represent the height in which different vertical sections of the flight have been combined to the respective shown profiles.”** *We have changed the color of the in-situ measurements to green and think the graphs are more distinguishable now. We changed furthermore the description of the figure: “The figure shows profiles of the PNC (solid line; shown on the left side of each panel) and CCN-NC (shown on the right) conducted during flight 14b (right panel) and 27a (left panel). The airborne in-situ measurements of the CCN are shown as dots and lidar-based measurements as a solid line with shaded area. Furthermore the ground-based measurements (red crosses, measured at the same time) of the respective parameters. Two airborne profiles p1 (black) and p2 (green) are shown for each flight. The shaded area around the lidar-based CCN-NC profiles symbolizes the uncertainty of factor two. Profiles p1 (black) and p2 (green) of flight 14b were recorded between 12.05 - 12.27 UTC and 13.47 - 13.54 UTC, respectively. The corresponding lidar-based profiles (lidar p1; black, and lidar p2; blue) were determined in the period 12.20 - 12.39 UTC and 13.15 - 13.29 UTC respectively. Profiles p1 (black) and p2 (green) of flight 27a were recorded at 10.08 - 10.34 UTC and 11.29 - 11.35 UTC, respectively. The corresponding lidar-based profiles were determined in the period 10.05 - 10.27 UTC and 11.25 - 11.57 UTC respectively. The horizontal black and blue dashed lines represent the height in which different vertical sections of the flight have been combined to the respective shown profiles.”*

*The updated version of the figure is given in Fig. R4.*

It would be worthy to include error bars, associated to the uncertainty of the measurements of the different variable, as the authors do for the case of lidar derived CCN.

*Author response: The uncertainty of the CCN measurements is about 10 per cent. In our opinion, the graphic with additional error bars would be too confusing. The uncertainties of the measurements are given in the manuscript.*

Concerning the uncertainties of lidar derived CCN the authors must explain what they are stating as "factor two".

*Author response: An uncertainty of factor two means an uncertainty range of half to double of the actual value. We added the red text to the manuscript on page 9 line 11: "factor two (uncertainty of half or double of the retrieved value; Mamouri and Ansmann, 2016)."*

Considering the discrepancy on CCN measured and derived from lidar it would be worthy to explore the use of different approaches for the lidar derived CCN, specially considering the features of the aerosol in the layers showing the largest discrepancy.

*Author response: Thank you very much for this idea. Nevertheless, we do not think that further studies of other approaches would lead to further results. On the one hand, the lidar approach has an uncertainty of at least factor two and thus the uncertainty range is already very large anyway. On the other hand, other approaches are based on the assumption of different aerosol types. Here we have not described the predominant aerosol type in more detail since it would be too speculative. However, it is possible that the aerosol of a height of about 1500 m may differ from the aerosol in layers below on September 27th. In any case, the backward trajectory on this day shows that the aerosol at around 1500 m altitude originates from other atmospheric layers than the aerosol at 1000 or 1500 m so that the shape of the PNSD and the composition is different in this altitude. In particular, in the approach of Mamouri and Ansmann (2016) they consider particles with a diameter of 100 nm as the reservoir for CCN in the case of continental aerosol. But, depending on the aerosol composition and type, the diameter from which particle serve as CCN can be smaller or larger so that the approach of Mamouri and Ansmann (2016) might under- or overestimates the CCN-NC. Furthermore, the parameters for the Eq. (8) ($c_{60,c}$ and $x_c$) are derived with a correlation of AERONET Sun-photometer products of the integrated particle number concentration $n_{50}$ (radius > 50 nm). These products are quite uncertain when the AOD is very low due to a very low aerosol load in the atmosphere similar to the conditions during flight 27a (cf. Fig. 10 - $N_{CPC}$ of ~2000 except the NPF events). The used parameterization therefore might be not applicable for the mentioned case. Possible reasons for differences of both approaches (in-situ and lidar) are mentioned in the manuscript.*

The discussion on the correlation between mCCN and lidar derived CCN must be corrected, as far as the logarithm of any variable have not units, the correct consideration would be to divide the CCN by 1 cm-3 and then compute the logarithm of a unitless variable. The axis in Fig 12 must be appropriately corrected. So the right comment must be: "On the second glance, in Figure 12 its clearly visible, that the lidar approach overestimates the airborne CCN-NC measurements for values of $\log_{10}(N_{CCN,mCCNc})$ from 2.7 to 3.4 ($N_{CCN,mCCNc}$ in the ranges 500 to 2500 cm$^{-3}$) by a factor of two, whereas in the range from $\log_{10}(N_{CCN,mCCNc})$ = 1.8 to 2.5 ($N_{CCN,mCCNc}$ 60 to ~320 cm$^{-3}$) the lidar approach underestimates. This indicates different aerosol types and explains the low correlation. Note, in the regime up to $\log_{10}(N_{CCN,mCCNc})$ = 1.8 the lidar approach.".

*Author response: We changed the axis labeling in Fig. 12. Furthermore we removed the units in the text (page 15, line 21-29) according to the suggestion. Updated version of the figure is given in Fig. R5.*

Considering the procedure followed in the derivation of the extinction profile from the backscattering profile using a fixed lidar ratio the comment: "The lidar profiles of $\sigma_{bsc}$ and $\sigma_{ext}$ show the same behavior" is rather obvious and can be omitted.

*Author response: We removed the mentioned sentence from the manuscript.*

In Tables B1 and B2 and in the discussion on LR just indicate the uncertainty with nor more than two significant figures (one if the most significant one is larger than 3) and then express the value plus minus uncertainty appropriately. For example:

68.84±10.33 must be 69+-10

23.21±3.48 must be 23+-3

153.75±23.06 must be 153+-23

*Author response: We adjusted Tab. B1 and B2 with the following function in Excel:*

*x=ROUND(value,digits-(1+INT(LOG10(ABS(value)))))*

*The values were either the value or the uncertainty and digits was 3 in the case of values and 2 in the case of uncertainty. Unfortunately we were not sure what the reviewer means with "one if the most significant one is larger than 3". The adjustments extended the table beyond page width so that the page format was changed to landscape mode.*

Just correct the small number of typos along the text.

*Author response: We found some typos and for instance we replaced "representativity" with "representativeness". It might be the case that we did not found all of the typos.*

The quality of some figures must be improved, specially the labels of Figures 14 -16.

*Author response: We adjusted the labeling of the legends of Fig. 14 – 16. Especially we increased the font size and made the legend for the fits simpler. For this purpose we included in the caption of the figure 14 the following sentence: "The colored lines represent the linear correlation of both parameters, with a the slope of the fit and R² the correlation coefficient. The black one is the 1:1 line. Filled symbols indicate data points determined within the planetary boundary layer and whereas empty symbols indicate data points above. Circles represent data points determined during flight 14a, triangles indicate flight 14b and squares 27a." The updated versions of the figures are given in Fig. R1, R2, and R3.*

*Updated Figures:*

[Figure]

**Figure R1: Scatterplot of the airborne-based ($\sigma_{bsc,mie}$) and the lidar-based ($\sigma_{bsc,lid}$) particle light backscatter coefficient for all horizontal legs during the investigated days for wavelengths λ = 355 (blue), 532 (green) and 1064 nm (red). The error-bars represent the assumed 15 %-error for the lidar-measurements and the three times standard-deviation of mean of the result of the Mie-calculations. The colored lines represent the linear correlation of both parameters, with _a_ the slope of the fit and _R²_ the correlation coefficient. The black one is the 1:1 line. Filled symbols indicate data points determined within the planetary boundary layer and whereas empty symbols indicate data points above. Circles represent data points determined during flight 14a, triangles indicate flight 14b and squares 27a.**

[Figure]

**Figure R2: Scatterplot of the airborne-based ($\sigma_{ext,mie}$) and the lidar-based ($\sigma_{ext,lid}$) particle light extinction coefficient for all horizontal legs during the investigated days for the wavelengths λ = 355 (blue) and 532 nm (green). $\sigma_{ext,lid}$ derived with a LR of 55 sr. The error-bars represent the assumed 15 % - error for the lidar-measurements and the three times standard-deviation of mean of the result of the Mie-calculations. Lines and symbols as in Fig. R2.**

[Figure]

**Figure R3: Scatterplot of particle light extinction coefficient derived with Mie-calculations ($\sigma_{ext,mie}$) and lidar-based ($\sigma_{ext,lid}$) for all horizontal legs during the investigated days for λ = 1064 nm. $\sigma_{ext,lid}$ derived with a lidar ratio (*LR*) of 30 sr. The error-bars represent the assumed 15%-error for the lidar-measurements and the three times standard-deviation of mean of the result of the Mie-calculations. Lines and symbols as in Fig. R1 and R2.**

[Figure]

**Figure R4: The figure shows profiles of the PNC (solid line; shown on the left side of each panel) and CCN-NC (shown on the right) conducted during flight 14b (right panel) and 27a (left panel). The airborne in-situ measurements of the CCN are shown as dots and lidar-based measurements as a solid line with shaded area. Furthermore the ground-based measurements (red crosses, measured at the same time) of the respective parameters. Two airborne profiles p1 (black) and p2 (green) are shown for each flight. The shaded area around the lidar-based CCN-NC profiles symbolizes the uncertainty of factor two. Profiles p1 (black) and p2 (green) of flight 14b were recorded between 12.05 - 12.27 UTC and 13.47 - 13.54 UTC, respectively. The corresponding lidar-based profiles (lidar p1; black, and lidar p2; blue) were determined in the period 12.20 - 12.39 UTC and 13.15 - 13.29 UTC respectively. Profiles p1 (black) and p2 (green) of flight 27a were recorded at 10.08 - 10.34 UTC and 11.29 - 11.35 UTC, respectively. The corresponding lidar-based profiles were determined in the period 10.05 - 10.27 UTC and 11.25 - 11.57 UTC respectively. The horizontal black and blue dashed lines represent the height in which different vertical sections of the flight have been combined to the respective shown profiles.**

[Figure]

**Figure R5: Correlation of the logarithmized (base 10) CCN-NC derived with the approach of Mamouri and Ansmann (2016) ($N_{CCN,lid}$) and directly measured with the mCCNc on ACTOS ($N_{CCN,mCCNc}$) for six profiles conducted during three flights (14a, 14b and 27a). Each profile has its associated lidar profile. Red line represents the line of fit and the black line the 1:1 line. Error bars represent the uncertainty of the lidar-based approach of a factor two and the 10 % uncertainty of the mCCNc on ACTOS.**